# Using computational fluid dynamics and field experiments to improve vehicle-based wind measurements for environmental monitoring.

Tara Hanlon[1] and Dave Risk[1]

[1]Department of Earth Sciences, St. Francis Xavier University, Antigonish, Nova Scotia, B2G 2W5, Canada

**Correspondence:** Tara Hanlon (thanlon@stfx.ca)

**Abstract.** Vehicle-based measurements of wind speed and direction are presently used for a range of applications, including gas plume detection. Many applications use mobile wind measurements without knowledge of the limitations and accuracy of the mobile measurement system. Our research objective for this field-simulation study was to understand how anemometer placement and the vehicle's external air flow field affect measurement accuracy of vehicle-mounted anemometers. Computational Fluid Dynamic (CFD) simulations were generated in Ansys FLUENT to model the external flow field of a research truck under varying vehicle speed and wind yaw angle. The CFD simulations provided a quantitative description of fluid flow surrounding the vehicle, and demonstrated that the change in windspeed magnitude from the inlet increased as the wind yaw angle between the inlet and the vehicle's longitudinal axis increased. The CFD results were used to develop empirical speed correction factors at specified yaw angles, and to derive an aerodynamics-based correction function calibrated for wind yaw angle and anemometer placement. For comparison with CFD, we designed field tests on a square, 12.8 km route in flat, treeless terrain with stationary sonic anemometers positioned at each corner. The route was driven in replicate under varying wind conditions and vehicle speeds. The vehicle-based anemometer measurements were corrected to remove the vehicle speed and course vector. From the field trials, we observed that vehicle-based windspeed measurements differed in average magnitude in each of the upwind, downwind, and crosswind directions. The difference from stationary anemometers increased as the yaw angle between the wind direction and the truck's longitudinal axis increased, confirming the vehicle's impact on the surrounding flow field and validating the trends in CFD. To further explore the accuracy of CFD, we applied the function derived from the simulations to the field data, and again compared these with stationary measurements. From this study, we were able to make recommendations for anemometer placement, demonstrate the importance of applying aerodynamics-based correction factors to vehicle-based wind measurements, and identify ways to improve the empirical aerodynamic-based correction factors.

# 1   Introduction

Many scientific applications require local measurements of wind speed and direction in the lower atmosphere. Currently, vehicle-based wind measurements are used to study severe weather-related meteorology (Belušić et al., 2014; Straka et al., 1996; Taylor et al., 2011), lake meteorology (Brook et al., 2013; Curry et al., 2017), and are integrated into methane measurement studies to detect, quantify, and map emission plumes from oil and gas developments (Atherton et al., 2017; Rella et al., 2015; Zazzeri et al., 2015).

Existing mobile measurement platforms include car, sport utility vehicle (SUV), and truck-mounted anemometers, although the quality and accuracy of the anemometer measurements is not well understood. As a vehicle travels, its motion creates an air flow field that is unique to the shape and velocity of the vehicle. To avoid measurement bias, instruments measuring wind speed and direction must be placed in a location that is not directly impacted by the flow and pressure perturbation produced by the moving vehicle (Straka et al., 1996). Mobile platforms often mount sensors in locations ahead of or above the vehicle (Atherton et al., 2017; Belušić et al., 2014; Raab and Mayr, 2008; Rella et al., 2015; Zazzeri et al., 2015). In the development of a mobile mesonet fleet (Straka et al., 1996), the importance of placing wind sensors outside of the vehicle's flow field was referenced, and the authors obtained wind tunnel tests from Nissan to determine the sensor placement. Another study (Raab and Mayr, 2008) mounted an anemometer atop a car, referencing that Computational Fluid Dynamic (CFD) analysis performed by car manufacturers showed the flow disturbance caused by a car was minimal near the car's frontal axle, at a height equivalent to one metre above the roof. These studies provide reasoning for sensor placement, but lack empirical study or simulation and quantitative understanding of the flow field to confirm that the measurements were indeed not impacted by the vehicle's flow and pressure perturbation. Smart sensor placement will reduce measurement bias, but measurements should also be calibrated for the bias to generate better accuracy, and greater certainty, of vehicle-based wind measurements.

The majority of studies evaluating the external flow field of vehicles do so for the purpose of evaluating aerodynamic drag (Yang and Khalighi, 2005; Holloway et al., 2009). In these studies, the spatial resolution around the vehicle is too limited to represent the complex flow field detail, and areas of flow separation cannot be identified (Defraeye et al., 2010). For the purpose of evaluating bias of a wind sensor measurement atop the vehicle, improved resolution is needed. Houston et al. (2016) recognized this problem, and created CFD simulations to evaluate the wind field at the location of wind sensors mounted atop a Dodge Caravan, finding that the vehicle caused the windspeed at the sensor locations to be overestimated by 4% in head wind, and could exceed overestimations of 9% in cross wind. Marine researchers have already conducted similar experiments, which led to correction functions for shipboard anemometers (Moat et al., 2005; Yelland et al., 1998). The shape of the ship's hull and the oscillating motion of the vessel distort airflow, causing a bias in wind measurements that is unique to the location of the anemometer (Moat et al., 2005). CFD and wind tunnel studies on Canadian research ships show that the anemometers placed ahead of the bow underestimate windspeed, whereas anemometers mounted on the main mast overestimate the windspeed (Moat et al., 2005). Wind tunnel studies show that the bow-positioned anemometers underestimate windspeed by a magnitude of three to five percent, while those at the main mast overestimate windspeed by a magnitude of five to ten percent (Thiebaux, 1990). In comparison, CFD studies indicate that anemometers positioned on the bow may decelerate the flow between 0 and

15 percent, while those at the main mast overestimate by approximately 5 percent (Yelland et al., 1998). Similar experiments have not been conducted for truck-based anemometers.

In this study, we set out to quantify truck-based anemometer measurement bias, and to derive an integrative placement and empirical calibration solution for vehicle-based anemometer measurements on a specific research vehicle. We used synthetic data from computational fluid dynamic simulations to obtain a spatial description of the behaviour of the flow field surrounding the vehicle, over a large range of vehicle speeds and wind yaw angles. The synthetically derived corrections were compared with field measurements acquired using a truck-mounted anemometer at multiple placements, with varying truck speeds and wind yaw angles. By evaluating the vehicle's external flow field under both varying vehicle speed and wind yaw angle we can quantify the bias the vehicle shape induces on a truck mounted anemometer and calibrate the measurements prior to correcting wind speed and direction measurements for the vehicle's motion.

## 2 Methods

### 2.1 CFD Study

CFD simulations were used to develop a quantitative understanding of fluid flow around the vehicle, and to evaluate flow under a wider range of conditions than is feasible in a field study. These data were used to investigate how the shape of the vehicle impacted measurement accuracy when the vehicle was travelling in the upwind and crosswind directions. We defined measurement bias as the difference between the inlet velocity and the velocity magnitude at the anemometer location. We created two sets of simulations in ANSYS FLUENT, the first of which was designed to evaluate the flow field under varying vehicle speed, and the second to explore the flow field under varying wind yaw angles.

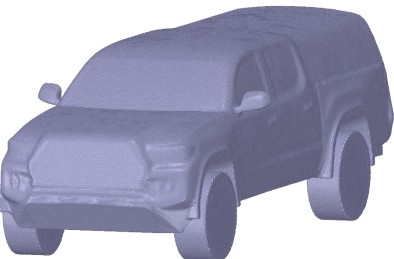

**Figure 1.** Truck model geometry replicating a 2016 Toyota Tacoma field vehicle.

The vehicle geometry was that of a 2016 Toyota Tacoma model equipped with a bed cap cover, similar to that used in Atherton et al. (2017), and modified to have a smooth underbody with simplified tires. The vehicle model (Fig. 1) was enclosed in a virtual wind tunnel extending 25.4 m above, ahead, behind, left, and right of the truck model. The computational domain

of the wind tunnel was 56.2 m in length, 53.0 m in width, and 28.2 m in height, with just under 84 000 m$^3$ of air volume. The wind tunnel was designed to have a large width so that the vehicle speed and yaw angle could be evaluated in a consistent manner across simulations. A fine resolution tetrahedron assembly mesh was generated in the ANSYS workbench package with 4.6 million cells, ranging from fine-scale adjacent to the surface of the truck to coarse farther away. The computational

wind tunnel was defined to have a velocity based inlet (1% turbulence intensity), and a pressure based outlet. We modelled the road and truck surfaces as stationary walls with no slip condition, and the top and sides of the tunnel with symmetry boundary conditions.

A drag coefficient comparison with the manufacturer's specification was used to compare the performance of three turbulence models. We calculated the drag coefficient of the truck model using a realizable two equation k -$\varepsilon$ model with non-equilibrium

wall functions, a two equation k-$\omega$ shear stress transport (SST) model, and a four equation transition SST model. All turbulence models are built into the ANSYS software and based on the steady state formulation, Reynolds Average Navier Stokes (RANS) equations. In simulations comparing the turbulence models, the drag coefficient was calculated using an inlet velocity of 22.2 m s$^{-1}$ (80 km h$^{-1}$), and the vehicle's projected frontal area of 2.819 m$^2$. The drag coefficient was determined when the change in C$_D$ was less than 0.001. The $\Delta$C$_D$ column was determined by comparing the calculated drag with the 2016 4x4 Double

Cab Toyota Tacoma specification value for C$_D$ of 0.386 (Toyota USA, 2016). The results of the C$_D$ comparison are displayed in Table 1. All turbulence models produced drag coefficients greater than the manufacturer's reported value. The addition of the cap on the truck model may have increased the frontal area, resulting in this slightly larger drag coefficient. The k -$\varepsilon$ model provided a C$_D$ value within 6% of Toyota's specification value, and was chosen as the most suitable model for our computational mesh.

Previous studies (Holloway et al., 2009; Roy and Srinivasan, 2000; Yang and Khalighi, 2005) have used k -$\varepsilon$ models for the purpose of external vehicle aerodynamics. The k -$\varepsilon$ models use two additional equations to solve the RANS equations, one for turbulent kinetic energy (k) and one for the turbulent kinetic energy dissipation rate ($\varepsilon$) (Roy and Srinivasan, 2000). The realizable k -$\varepsilon$ model with non-equilibrium wall functions was advantageous for this study. Non-equilibrium wall functions are sensitized to adverse pressure gradients and predict flow behaviour in turbulent boundary layers better than the traditional k -$\varepsilon$

model, which is limited in cases of larger adverse pressure gradients (Parab et al., 2014). The k -$\varepsilon$ model has been previously used to study external vehicle aerodynamics. Holloway et al. (2009) compared a steady k -$\varepsilon$ model with two transient turbulence models, concluding that all of the models capture the general trends in the flow field aft of the cab of a pickup truck. In a study comparing CFD and experimental data, similar flow structures were observed from a steady k -$\varepsilon$ CFD model and time averaged wind tunnel test experiments when comparing velocity planes parallel to the ground (Yang and Khalighi, 2005). The pickup

truck models in both these studies had an open bed. In our study, the model we were exploring has a cap on the bed. The flow field around a pickup truck is more complex than the flow around an SUV or sedan because of wake interactions (Yang and Khalighi, 2005). The cap on the truck bed eliminates the pressure drop that occurs on pickup trucks when the air flows over the cabin and into the bed. Our motivation for placing an anemometer atop a pickup is to assist vehicle-based gas monitoring systems (Atherton et al., 2017; Baillie et al., 2019; O'Connell et al., 2019) measuring gas emissions to achieve practical

placement and further calibration practices. In these studies, the anemometer was mounted above a truck cap. Because of the

**Table 1.** Drag comparison of three turbulence models with inlet velocity of 22.2 m s$^{-1}$. The drag coefficient (C$_D$) for each turbulence model is reported with the number of iterations required to achieve a solution and the percent difference ($\Delta$C$_D$) from the manufacturers reported (C$_D$) of the vehicle model.

| Turbulence Model | Iterations | C$_D$ | $\Delta$ C$_D$ |
|---|---|---|---|
| k-$\varepsilon$ | 436 | 0.411 | 6 % |
| k-$\omega$ | 234 | 0.417 | 8 % |
| SST | 217 | 0.446 | 15 % |

differing vehicle shape from an open bed pickup truck, k -$\varepsilon$ performance in the drag comparison, and performance in other studies, we have selected the k -$\varepsilon$ turbulence model as appropriate for our analysis in the velocity field above a capped pickup truck.

In the first set of simulations, the above model was used to investigate the flow over the vehicle at speeds ranging from 40 to 100 km h$^{-1}$ in 5 km h$^{-1}$ increments. All simulations were performed in ANSYS Fluent 17.2, and were run in parallel using 32 cores dispersed over two nodes using 2.2 GHz Operton or Intel Xeon 2.7 GHz clusters made available by Compute Canada.

In the second set of simulations, eight additional computational domains were created to explore the effects of wind yaw angle, each representing a different wind yaw angle for the truck. In each case, the truck's longitudinal axis was rotated so that the inlet was perpendicular to the yaw angle direction of flow. The computational domain was to made to extend the same 25.4 m in each direction as the original tunnel. The truck was rotated counterclockwise in 5° increments to provide models between 5° and 40°, inclusive. The symmetry plane of the truck was used to simulate the corresponding yaw angles of 320° to 355°, inclusive. We assumed that the wind atop the truck on the passenger side at 5° will represent the same wind that the driver side of the truck experiences if the truck was rotated 355°. We used the additional computational domains to test all combinations of yaw under inlet speeds ranging from 40 km h$^{-1}$ to 100 km h$^{-1}$ in 5 km h$^{-1}$ increments.

## 2.2 Field Measurements

The field experiments were designed to validate the CFD results, using stationary anemometer measurements as a control comparison for truck-based anemometer measurements. For comparison with CFD, it was necessary to obtain truck-based wind measurements from a range of yaw angles. Field tests were carried out using a 3.2 km by 3.2 km driving route, with a stationary anemometer positioned near each corner in order to compare truck-based anemometer measurements under head wind, tail wind, and side wind conditions, with those acquired by the stationary anemometers. The stationary anemometers were placed on the corners to provide opportunity for comparison with both legs of the straightaway. This ensured that we would have data for comparison with each leg should an anemometer malfunction. The square route also provided vehicle-based measurements for tail wind conditions, where stationary anemometers recorded wind direction opposing the vehicle path. The terrain was flat, treeless, nearly devoid of infrastructure, and we experienced minimal traffic.

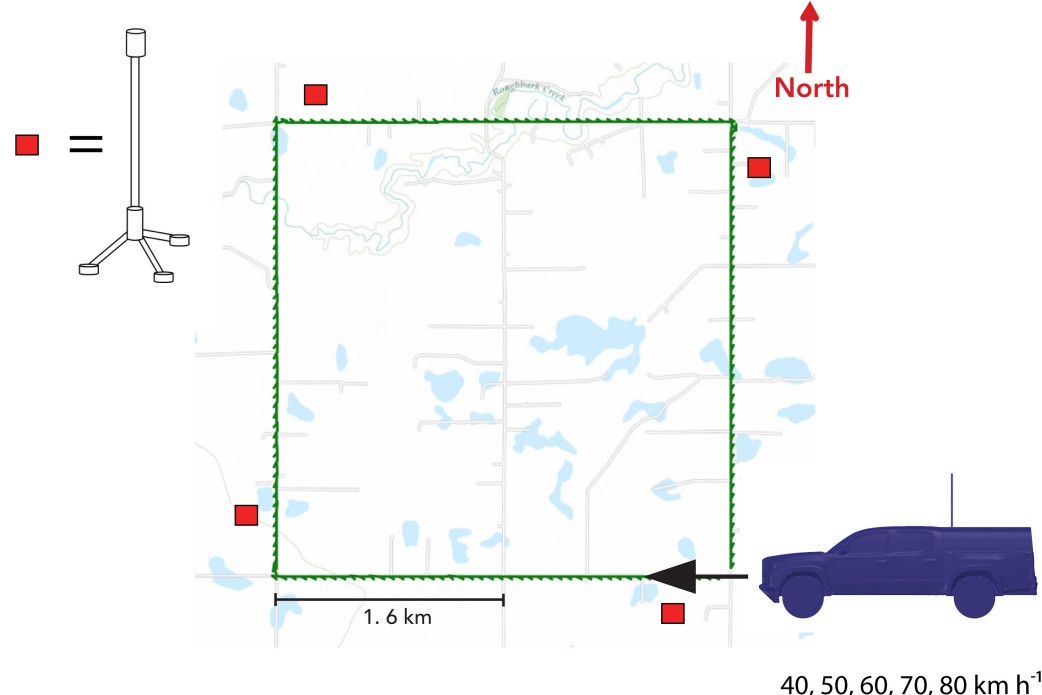

North

1. 6 km

40, 50, 60, 70, 80 km h$^{-1}$

**Figure 2.** Field schematic displaying the location of stationary anemometers, the site, and the vehicle path. The vehicle path was driven clockwise and repeated at speeds of 40, 50, 60, 70, and 80 km h$^{-1}$.

In summer 2017, we drove the route at speeds of 40, 50, 60, 70, and 80 km h$^{-1}$. Our hardware consisted of an RM Young 86004 ultrasonic anemometer mounted on a 1.2 m tall mast above the cap of a 2016 Toyota Tacoma, with a total height of 3 m above the surface. The anemometer was positioned 0.5 m from the longitudinal axis of the truck on the driver's side, and 0.3 m behind the start of the truck's cap, in the location of the roof racks. A Campbell Scientific CR1000x datalogger recorded time,

5    wind speed, wind direction, and instantaneous geolocation from a Garmin GPS 18X, at 1 Hz. Four stationary tripod-mounted Decagon Device DS-2 sonic anemometers measured wind speed and wind direction at a height of 3 m above the ground. The tripods were located approximately 300 m clockwise from each corner to allow for the truck to be travelling at a constant speed when passing the stationary instrument. Measurements of time, wind speed, maximum gust, and direction were recorded with a Decagon Device Em50 datalogger in one minute intervals at the same time we were driving the route. The Decagon

10    anemometer model was selected based on cost effectiveness, and the RM Young model was selected for its ability to measure high wind speeds, necessary for the mobile wind measurements. A schematic describing the field experiment is displayed in Fig. 2. The calculated vehicle bearing measurements showed that the roads used in the square route were within 2° of North, South, East and West directions.

To explore the effects of the anemometer height above the truck, we repeated tests with the addition of a secondary anemometer at a lower position of 0.3 m above the truck (2.1 m above the ground), and 0.4 m from the longitudinal axis of the truck on the driver's side.

The 1Hz geolocation measurements were used to compute the vehicle speed and bearing, field measurements that deviated more than 2.5 km h$^{-1}$ from the implemented cruise control speed, vehicle bearing measurements that differed more than 5° from designated vehicle course were removed from the data set. The vehicle anemometer measurements from each field test were corrected to remove the vector of the vehicle's motion. To correctly compute true winds, vehicle based wind must be corrected for the vehicle's motion over the fixed earth. The vehicle vector has a direction equivalent to the vehicle's course over the ground, and a magnitude equivalent to the vehicle's speed over the ground (Smith et al., 1999). The 1 Hz wind direction and wind speed measurements from the truck anemometer were each used to create the real and imaginary components for the wind vector, with the coordinate system aligned so that the front of the truck was 0 degrees. The wind vector was translated to match the truck's coordinate system, with North as 0 degrees. We calculated the vehicle's course over the ground using the 1 Hz GPS coordinates to obtain a vector of equal magnitude and opposite direction to the frontal wind induced by the vehicle's motion (F). We compute the meteorological wind vector (W) by subtracting the frontal wind induced by the vehicle's motion (F) from the raw anemometer measured wind vector (A). The calculation is presented in Equation (1) . The computation comes from the method calculating the meteorological true wind from a moving vessel presented by Smith et al. (1999).

$$W = A - F \tag{1}$$

## 3   Results

The CFD simulations and field experiments show that vehicle-based anemometers are subject to bias as a result of the vehicle's flow field. The CFD and field experiments showed that the measured wind at the anemometer location varied under wind yaw angle. The bias found in the CFD varied with the rotation of the truck, and the field results concluded that the measured wind differed in head, cross and tail wind conditions. We first present the CFD results, followed by those from the field.

### 3.1   CFD results

The CFD simulations provided a quantitative description of the air flow surrounding the vehicle. Velocity fields were the interest of this study, and we observed that wind yaw angle has a more pronounced effect on the wind speed bias than the vehicle speed does. This result was expected as vehicle's are designed to be aerodynamic in the forward direction of motion.

The first set of simulations was evaluated to explore the effect of vehicle speed on wind measurement bias. In each computational domain, we found that the wind bias scaled with vehicle speed and the amount of bias (slope) differed with the location above the vehicle. Similarly, Houston et al. (2016) found that CFD calculated velocities for sensors positioned atop a van scaled linearly with along axis speed. Our location of interest for placing anemometers was on the truck's bed cap. Velocity contours

were computed for the longitudinal axis of the truck, and for the lateral plane on the truck located 30 cm from the end of the cab, which was the location of the roof racks on which the anemometers were mounted in the field test.

Figure 3 displays a velocity contour for the overestimation of the speed along the longitudinal axis of the truck. In this plane, we found that the velocity flowing over the truck was less than the inlet speed at small heights above the truck cabin and cap. In the location of the roof racks, the flow velocity was less than the inlet speed at heights lower than 17 cm above the truck. At 17 cm, the velocity flow transitioned to become larger than the inlet speed. The velocity gradient with respect to height was largest in the region where the flow transitioned to becoming larger than the normalized wind speed, and decreased with increasing height. The magnitude of flow acceleration was greatest immediately above 17 cm, and decreased with increasing height above the truck. This observation is similar to a conclusion by Moat et al. (2005), stating that shipboard anemometers should not be placed close to the line of equality (where measured windspeed = true windspeed) as large pressure gradients are present in this region. The line of equality would be expected to move vertically to some degree, according to the ground speed and/or windspeed.

The second set of simulations in this study are of particular interest because they evaluate the flow field under cross wind conditions. Many previous CFD and wind tunnel studies (Holloway et al., 2009; Yang and Khalighi, 2005) examine the flow field resulting from inlets which direct the airflow along the longitudinal axis of the vehicle. Aside from Houston et al. (2016), we have not found studies using CFD to conduct crosswind experiments to study the flow field around a vehicle. In performing this analysis, we observed that the change in windspeed magnitude from the inlet increased along with the wind yaw angle between the inlet and the vehicle's longitudinal axis. The flow above the truck displayed a profile where decelerating flow was found below a region of accelerating flow before reaching the undisturbed velocity flow. The maximum magnitude of flow acceleration that occurs above the deceleration region increased with increasing yaw angle. Figure 4 shows the profile of velocity contours in the lateral plane of the truck, along the location of the roof racks on the truck when the truck is exposed to frontal, 15° passenger, and 30° passenger wind. When exposed to large yaw angles, the wind bias over the truck at low heights can be twenty percent, and even at a height of 1.7 m above the vehicle, a bias of greater than 5 percent is present.

Anemometers must be placed away from the region of sharp velocity gradients to avoid wind bias. The height of this region changes based on the angle at which the wind is flowing over the truck. From the velocity profiles, we conclude that anemometers measuring wind speed and direction must be mounted at a significant height above the vehicle. For the velocity profile of the longitudinal axis of the truck, we found that the flow differential dropped below 2% at a height of 2.59 m above the truck and below 1% of the inlet speed at a height of 4.36 m above the truck. These heights are displayed in Fig. 3. Looking at the longitudinal velocity profile located 50 cm left of the truck's axis, we found that the flow acceleration dropped below 2% of the inlet speed at a height of 2.61 m above the truck, and below 1% of the inlet speed at a height of 4.47 m above the truck. From this, we can conclude that an anemometer mounted off the centre line must be positioned higher than one mounted in the centre of the truck. Furthermore, we found that the wind yaw angle was critical for determining the height at which to mount a sensor atop a vehicle. Table 2 shows the minimum height required to mount a sensor where the flow acceleration is less than 2% for an anemometer mounted on the longitudinal axis of the truck, 0.3 m behind the truck cab. It is expected that the bias of

less than 2% would fall within sensor accuracy. The RM Young anemometer used in our field experiment had an accuracy of 2% for speeds of less than 30 m/s, and 3% for speeds between 30 and 70 m/s.

Anemometer placement should not be determined from wind tunnel tests with directly frontal flow. Table 2 shows that the height required for an anemometer to experience bias below 2% increases with increasing yaw angle. Anemometer heights selected through frontal wind tunnel tests would still experience bias under cross wind conditions. It is expected that vehicle-mounted anemometers would be subject to yaw angles between 0° and 40°. For perspective, if a vehicle was driving at 80 km h$^{-1}$ perpendicular to a wind of 22 km h$^{-1}$, the apparent yaw is expected to be 15 degrees. In days with high winds and low driving speeds, it is possible to experience a yaw angle of 40 degrees. The bias experienced at the anemometer height in our field tests is greater than the instrument accuracy, therefore we must correct anemometer measurements for flow distortion.

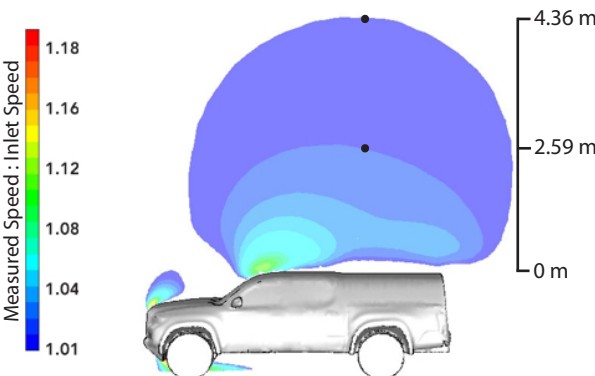

**Figure 3.** Frontal wind velocity contour of the truck's longitudinal axis. The black dot at 2.59 m represents the anemometer height required to be subject to less than 2% bias, and the black dot at 4.36 m represents the anemometer height required to be subject to less than 1 % bias, if mounted 30 cm behind the truck cab.

We derived aerodynamic-based correction factors to be applied to vehicle-based wind measurements from the simulation datasets. The normalized wind speed ($\dfrac{\text{measured wind speed}}{\text{inlet speed}}$) at the location of an anemometer mounted in the same location as the field test was computed for each yaw angle simulation. The empirical correction factor for each yaw angle is the reciprocal of the normalized windspeed. Empirical correction factors were computed for the anemometer placement in the field trials, and for an anemometer placement in the centre of the truck, above the roof racks. The correction factors for both anemometer placements were fitted with weighted polynomial regressions. Figure 5 shows the function for both placements. The centred placement provides a symmetrical function, whereas the side-mounted anemometer measures lower windspeed coming from the driver side than the passenger side. The wind coming from yaw angles over the passenger side experiences similar bias in both placements. We conclude that, by moving the anemometer to the side of the truck, we reduce the bias from wind yaw angles experienced on the driver side. Figure 5 shows that it is important that the anemometer correction function is calibrated for the anemometer placement. The polynomial pictured in Fig 5 is multiplied by the Anemometer windspeed ($A_{WS}$) to give

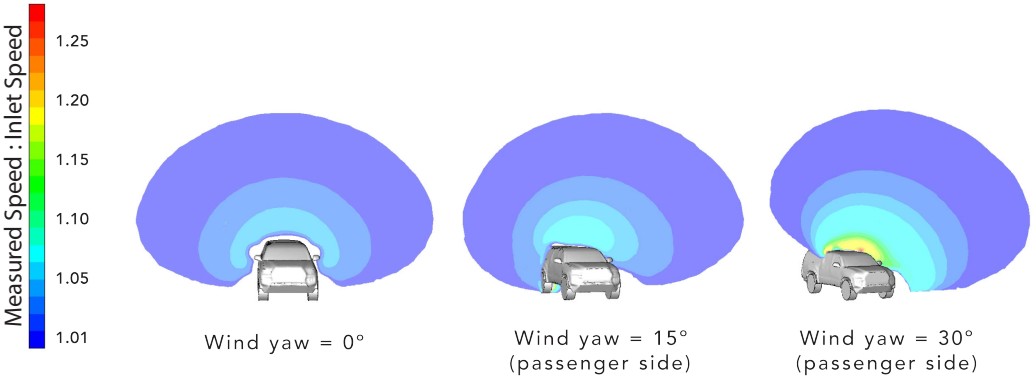

**Figure 4.** Velocity contours of the lateral plane of the truck along the roof racks for yaw angles of $0°$, $15°$, and $30°$. Yellow regions indicate that the wind is accelerated by 18%.

**Table 2.** Required anemometer height for a centrally-mounted anemometer to experience negligible bias.

| Yaw Angle | Height |
|-----------|--------|
| 0° | 2.59 |
| 5 ° | 2.67 |
| 10° | 2.87 |
| 15° | 3.01 |
| 20° | 3.28 |
| 25° | 3.48 |
| 30° | 3.73 |
| 35° | 3.92 |
| 40° | 4.04 |

the corrected wind speed (WS). Equation 2 gives the side-mounted anemometer's correction function for wind direction (WD) measurements ranging from $-40° <$ WD $< 40°$, where the wind direction measurement is measured in the truck's coordinate system with $0°$ facing the front of the truck. Wind speed units are in km h$^{-1}$ and wind direction units are in degrees.

$$WS = A_{\text{WS}} \times \left( 9.6286(10^{-1}) - 1.4166(10^{-4})WD - 5.5849(10^{-5})WD^2 + 9.7413(10^{-8})WD^3 + 1.5485(10^{-8})WD^4 \right) \qquad (2)$$

5    The side-mounted anemometer placement was used in our field tests. For our field analysis, we will use Eq.(2) as an empirical correction function calibrated for anemometer placement and wind yaw angle. The empirical calibration must be applied to remove the bias of the vehicle's shape on anemometer measurements prior to correcting for the vehicle motion.

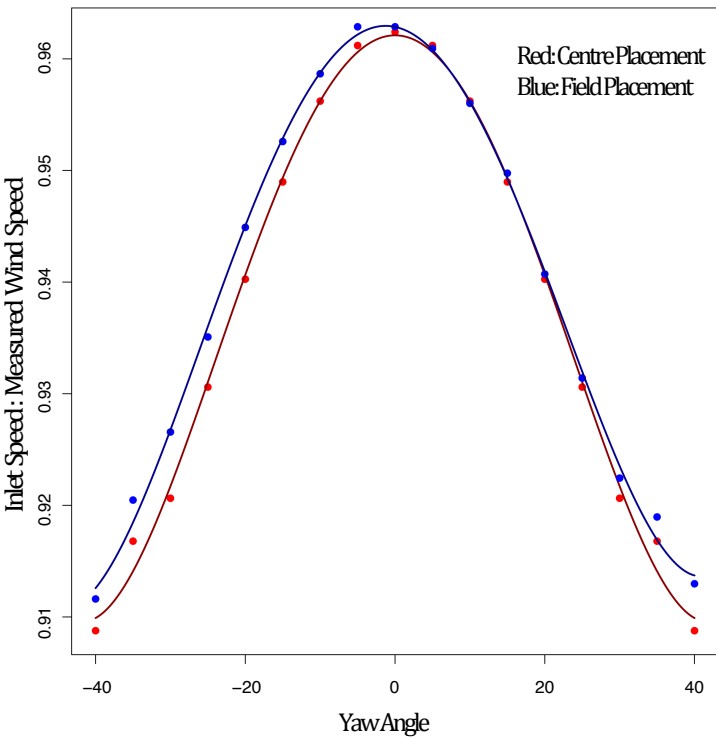

**Figure 5.** Empirical correction factors for field placement (blue) and centred placement (red) fitted with weighted polynomial regression.

## 3.2 Field Results

The field results provided measurements of the flow field at two locations above the truck. While the spatial resolution of the flow field was limited, the measurements were able to provide data true to the application. Vehicle-based wind measurements were acquired on separate days in predominantly North, East, South, and West winds, and with the average wind speed of the field tests ranging from 13.5 km h

To evaluate the vertical velocity profile, we compared the measurements from the two anemometer tests. The short anemometer measured larger mean wind speeds in the head, cross, and tail wind directions, demonstrating that vehicle speed had a greater impact on the windspeed measurements of the short anemometer. Figure **??** shows the magnitude of the tall and short

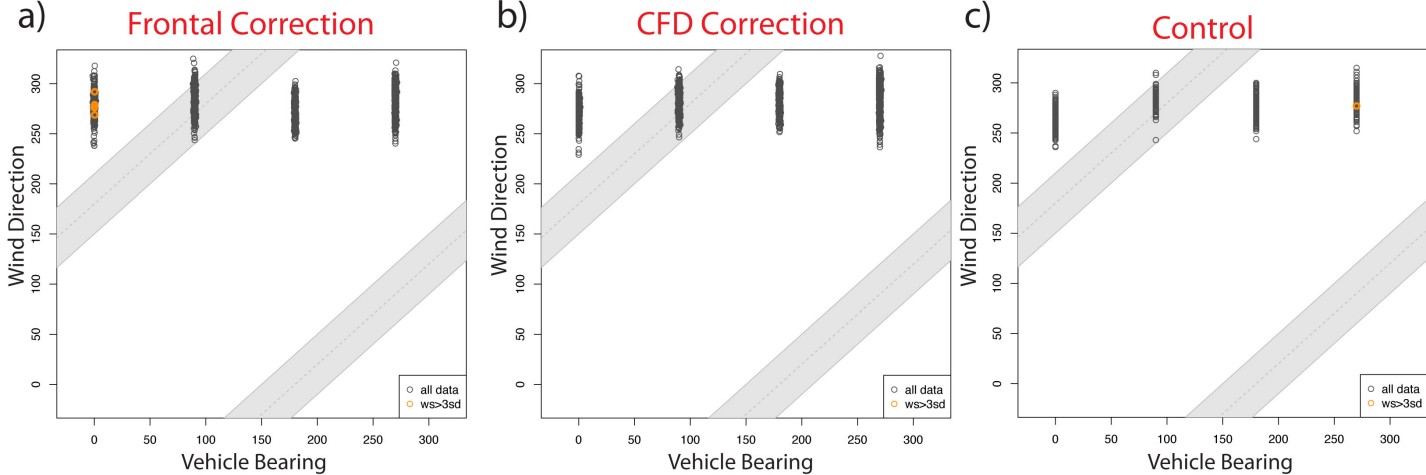

**Figure 6.** Distribution of wind measurements across vehicle bearing with (left) frontal correction applied to mobile measurements, (centre) empirical correction and frontal correction applied to mobile measurements, and (right) stationary measurements. The shaded grey bars represent measurements acquired when the vehicle was driving in tail wind conditions. The yellow circles represent wind speed measurements that are greater than three standard deviations away from the mean wind speed of the wind speed measurements calculated with the a) frontal correction applied, b) empirical correction and frontal correction applied, and c) specific to the stationary anemometer at each bearing.

anemometer measurements when normalized by the vehicle speed. We concluded that vehicle speed had a larger effect on the short anemometer placement.

The field data were compared with the stationary anemometers. For our results, we present the corrected field data in two ways: (1) with the frontal wind correction, and (2) with the applied CFD empirical correction, and then the frontal wind correc-
5 tion. We found that the application of the frontal wind correction overestimates windspeed in head wind, underestimates in tail wind, and differs in passenger side and driver side wind. Applying the CFD empirical correction reduces the overestimation in headwind and the underestimation in tail wind. It also improves the windspeed in passenger and driver side wind.

We found that applying the empirical correction reduces the windspeed measurement outliers, and reduces the standard deviation of the wind direction measurements. Figure 6 shows the distribution of wind measurements across vehicle bearing.
10 Figure 6 displays field test data with the frontal wind correction (left), the field data with the empirical correction applied, followed by the frontal correction (centre), and the stationary measurements, with the vehicle bearing being that which the vehicle was driving when adjacent to that anemometer (right). Applying the empirical correction reduces the difference in the mean of the wind direction measurements across the four vehicle bearings. The plots outline the areas the vehicle was driving in tail wind, as we had expected more outliers to occur when driving in tail wind. The control plot shows that outliers can be
15 expected due to the natural variability of the wind.

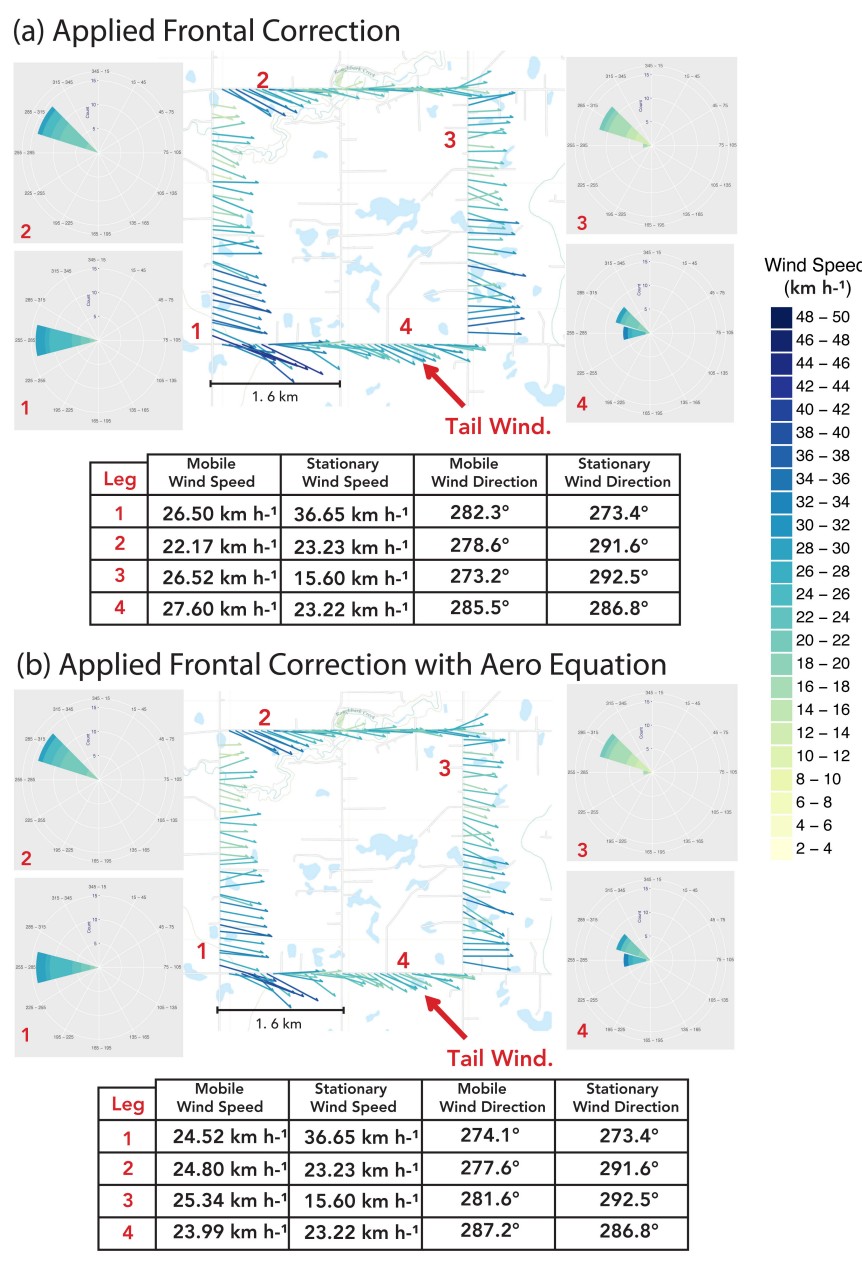

## (a) Applied Frontal Correction

| Leg | Mobile Wind Speed | Stationary Wind Speed | Mobile Wind Direction | Stationary Wind Direction |
|-----|-------------------|-----------------------|-----------------------|---------------------------|
| 1 | 26.50 km h-1 | 36.65 km h-1 | 282.3° | 273.4° |
| 2 | 22.17 km h-1 | 23.23 km h-1 | 278.6° | 291.6° |
| 3 | 26.52 km h-1 | 15.60 km h-1 | 273.2° | 292.5° |
| 4 | 27.60 km h-1 | 23.22 km h-1 | 285.5° | 286.8° |

## (b) Applied Frontal Correction with Aero Equation

| Leg | Mobile Wind Speed | Stationary Wind Speed | Mobile Wind Direction | Stationary Wind Direction |
|-----|-------------------|-----------------------|-----------------------|---------------------------|
| 1 | 24.52 km h-1 | 36.65 km h-1 | 274.1° | 273.4° |
| 2 | 24.80 km h-1 | 23.23 km h-1 | 277.6° | 291.6° |
| 3 | 25.34 km h-1 | 15.60 km h-1 | 281.6° | 292.5° |
| 4 | 23.99 km h-1 | 23.22 km h-1 | 287.2° | 286.8° |

**Figure 7.** Mobile wind vectors displayed with adjacent stationary wind roses with a) frontal correction applied to mobile measurements, and b) empirical correction and frontal correction applied to mobile measurements. The vehicle travelled this route in a clockwise direction. The wind roses show the direction from which the wind is blowing, the mapped vectors indicate the direction to which the wind is blowing. The table below each plot indicates the averaged stationary and mobile wind speed and wind direction for each leg.

Figure 8 shows mapped vehicle wind vectors under the frontal (Fig. 7a) and empirical and frontal (Fig. 7b) corrections. Wind roses from the stationary anemometers on each leg are located in the corners. The measurements in the wind roses are the wind gust and average wind direction measurements reported each minute of the 14 minute 60 km h$^{-1}$ speed test.

The mobile wind vector measurements in these plots are not averaged, and they were all taken when the vehicle was travelling within 2.5 km h$^{-1}$ of 60 km h$^{-1}$. Both plots look similar, with the largest change being the reduction of the speed of the arrows in the empirical correction plot. Figure 7 shows that applying the CFD calibration improves the agreement of the average mobile wind speed and direction measurements with the average stationary measurements in three of the four legs of the route. Observations of windspeed as-measured (uncorrected for flow) show that when the truck was moving faster than 40 km h$^{-1}$, 72% of raw wind direction measurements fell between -40 and 40 degrees of the front of the truck, the same as the yaw angles simulated in CFD. Field measurements were the only method of obtaining data under tail wind conditions in our study. Our CFD model was limited to a single inlet, and did not have the capability to simultaneously model oncoming air from the vehicle's motion, and wind blowing from behind the truck. Empirical correction factors could be derived from field experiments to provide better data for tail wind conditions. Our square experiment provided successful data for validation. The experiment could be expanded to include repetitive routes of each vehicle speed over a larger range of field days for exposure to more wind speeds. It would also be important to test on a day with little wind to identify a minimum wind speed to driving speed ratio for field practice. The resulting vehicle anemometer dataset with corresponding stationary measurements could be used to derive empirical correction factors. The data in our study only provided one square test per speed each day, and was unable to provide data at the range of yaw angles in which CFD was successful. However, square-route driving experiments would still be a good way to obtain a field-based empirical correction function, at least for the range of windspeed and wind yaw angle conditions experienced during the test.

We found that applying the CFD empirical correction function, followed by the frontal wind correction, reduced the number of wind speed outliers found on the square route. The plotted wind vectors in Fig. 8 show that the 1 Hz mobile wind measurements experience the most variability in speed and direction when the vehicle is travelling in tail wind conditions. The empirically corrected tail wind direction measurements were in reasonable (within ~30°) agreement with stationary direction measurements on field days with winds greater than 25 km h$^{-1}$, but deviated on days with lower wind speeds. When the vehicle speed is much larger than the wind speed, the vehicle creates a continuous flow field that can obstruct natural winds. The anemometer detects continuous frontal wind from the vehicle, and becomes less sensitive to components of the wind coming from behind the vehicle.? It is likely that a windspeed threshold for the magnitude of tail wind detected by the mobile anemometer, and that this threshold varies with vehicle speed. Additional field testing on field days with low winds is recommended to evaluate the quality of tail wind measurements, identify windspeed thresholds, and develop quality control criteria for tail wind measurements. As CFD is limited in developing an empirical correction for tail wind, additional filed data is recommended to develop a correction for tail wind measurements.

We applied the frontal wind correction to both mobile anemometers, and explored the difference in the mean windspeed at each vehicle speed in head, cross, and tail winds. The results are displayed in Table 3. The cross-wind measurements used in this test were from the passenger side of the truck, as the post of the tall anemometer may have impacted the short anemometer

**Table 3.** Percent difference between mean short and tall anemometer measurements across vehicle speed.

| Speed (km h$^{-1}$) | Head Wind | Cross Wind | Tail Wind |
|---|---|---|---|
| 50 | 0.0% | 5.3% | 6.6% |
| 60 | 0.0% | 5.7% | 4.6% |
| 70 | 2.1% | 5.7% | 10.2% |
| 80 | 1.0% | 5.2% | 3.3% |

measurements when the wind was coming from the driver side. The percent difference scales well across vehicle speed in head and cross winds. The agreement of the field measurements across vehicle speed shows that the wind speed scales linearly with vehicle speed and validates the trend in our CFD model. The measurements deviate in tail wind.

## 4    Discussion

Wind bias from mobile anemometers could lead to volumetric error in methane emissions from oil and gas infrastructure. Plume dispersion applications feed wind measurements paired with gaseous concentrations from mobile measurement platforms into gaussian dispersion models to locate emitting infrastructure, estimate source emission rates, and quantify emitted volumes of methane in oil and gas developments (Atherton et al., 2017; Caulton et al., 2017). In the gaussian dispersion model, emission rate scales linearly with wind speed. Our CFD study has shown that the shape of the vehicle accelerates wind speeds between 3% and 10%, subject to wind yaw angle. When measuring downwind from infrastructure, the error in windspeed translates linearly to the error in calculated emission rate. Anemometer placement and measurement methodology should be assessed together to minimize potential wind bias prior to using wind measurements in dispersion models. Mobile surveying studies using trucks or sport utility vehicles (Atherton et al., 2017; Jackson et al., 2014; Phillips et al., 2013; Rella et al., 2015; Zazzeri et al., 2015) with anemometer placements above the vehicle are vulnerable to flow bias, and should use flow compensations to account for wind bias from the shape of the vehicle. Transect-based studies with anemometers placed atop sport utility vehicles (Caulton et al., 2017), should also apply flow compensations, although some transect-based studies using mobile laboratories (Roscioli et al., 2015; Yacovitch et al., 2015) with anemometers placed on a boom ahead of and above the vehicle are much more resilient to bias from the flow of the vehicle. Similarly, studies quantifying emissions in which the vehicle stops to obtain wind measurements (Brantley et al., 2014) and the anemometer is placed ahead of and above the vehicle, are unlikely to require compensations for the vehicle's flow field. Vehicles outfitted to study severe weather (Curry et al., 2017; Taylor et al., 2011), and urban meteorology (Joe et al., 2018) show resilience to bias by also outfitting vehicles with wind sensors above and ahead of the vehicle. Vehicle-based anemometer measurements should be calibrated for vehicle shape and anemometer placement but as this can be costly and time consuming truck-based anemometers should be placed as far forward and as high as possible to obtain the most accurate results.

The calibration of vehicle-based measurements is important for integration with stationary measurements, and with mobile measurements from differing vehicle platforms. Studies evaluating the trends in near-surface ocean winds demonstrated that

systematic bias in measurement methods contributed to an increasing trend in global windspeed in reported climate data (Cardone et al., 1990; Ramage, 1987; Peterson and Hasse, 1987). The International Comprehensive Ocean Atmosphere Data Sets (COADS) document wind measurements from Voluntary Observing Ships (VOS) and other marine platforms (Thomas et al., 2008). The datasets date back prior to the 1940s and provide data for observed changes in climate patterns (Ramage, 1987). The datasets have been studied extensively to explain the increasing trend in global wind speed after the 1940s (Cardone et al., 1990), when archived data prior to the 1940s showed a decreasing trend (Thomas et al., 2008). Peterson and Hasse (1987) and Ramage (1987) attributed the shift from decreasing to increasing wind speed to the change in the method of reporting ship-based wind measurements. The reported wind speeds range from estimated measurement based on sea state to recorded measurements from ship-mounted anemometers of varying height. The Beaufort Wind scale, a method for visually estimating the wind speed in relation to sea state characteristics, was introduced in 1946, and wind reports evolved from being derived from the amount of sail a ship could carry, to being derived from observation of sea state. The increasing trend after the 1940s has also been attributed to a change in measurement techniques, and to variation in anemometer height on ships. Peterson and Hasse (1987) found that as anemometers were introduced on research ships, the distribution of the reported Beaufort velocities changed significantly. The measurement of gust readings made available by anemometers influenced the estimation of the Beaufort force, as the values derived from sea state were reported to be higher. Thomas et al. (2008) attributed the gradual increase in average ship anemometer height as another contributor to the increase in mean windspeed. Thomas et al. (2005) compared wind reports from ships and buoys, and noted that ship winds were reported 25 % higher than buoy winds. Measured winds have subsequently been adjusted for height using a logarithmic profile, resulting in the measurements differing by only 6 %. While these adjustments were important, measured winds still are not calibrated for the acceleration or deceleration of flow in the anemometer location. Moat et al. (2005), Thiebaux (1990), and Yelland et al. (1998) indicated that ship-based anemometers are subject to bias between 0 and 15, with bow-placed anemometers on the lower end. Applying ship-based anemometer calibrations could further reduce the bias between ship and buoy winds.

Theoretically, vehicle-based measurements of wind speed and direction could be integrated with fixed site measurements to add spatial richness in climate, weather, and atmospheric observing systems. Our CFD results compare well with Houston et al. (2016) for sensor placements atop a vehicle, and suggest that flow compensations should be made for vehicle-based anemometers. Calibrating vehicle-based measurements for anemometer placement and vehicle shape make wind measurements comparable with adjusted weather station data, and can provide data to form an observing system of land-based surface winds. Vehicle-based wind measurements from field studies can be used to contribute to detailed observing networks of specific sites. Furthermore, vehicle-based wind measurements can be collected conveniently by vessels of opportunity that travel routine routes, and contribute to weather data assimilation to evaluate accuracy for air quality and weather forecasting models (Brook et al., 2013). To provide quality measurements, consistent processing techniques must also be developed to avoid systematic bias introduced by averaging and filtering.

## 5    Conclusions

Mobile measurement platforms are capable of providing spatial and temporal measurements of wind speed and direction. Vehicle-mounted anemometers are impacted by the vehicle's motion and the vehicle's flow field at the location of measurement. Increasing the height above the vehicle at which the sensor is mounted reduces the impact of the vehicle's flow field on measurements. Although the height required to completely eliminate the effect results in an impractically high position, empirical or CFD-derived corrections can help to reduce bias. For applications requiring near surface wind measurements to be paired with other vehicle-based measurements, and similar anemometer placements to those in this study, we recommend anemometers on truck caps be mounted at least 1 m above the vehicle, and that an empirical correction be applied. While mounting the anemometer at larger heights above the vehicle reduces the impact of flow distortion resulting from the vehicles motion, placement must be practical and safe for road travel. For larger vehicle's, we expect mounting anemometers on vehicle's at heights much greater than 1 m to be potentially impractical. It is important that placement keeps the sensor below common road clearance limits for bridges and underpasses. CFD and field methods are both appropriate methods for deriving corrections. The calibration of wind speed and direction measurements through CFD-derived empirical calibrations was shown to be effective in reducing the bias that results from the vehicle's air flow field. Field measurements with good control data can also provide datasets for developing calibrations.

*Data availability.*  TEXT

*Competing interests.*  The authors declare they have no conflict of interest.

*Acknowledgements.*  The authors would like to thank Nayani Jensen for her help in collecting field data. We would also like to thank Whitecap Energy staff and Andy Froncioni of Garmin International for their contributions that helped inspire and shape this study. This research was made possible with resources made available by ANSYS and Compute Canada.

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
