# Peer review of "Using computational fluid dynamics and field experiments to improve vehicle-based wind measurements for environmental monitoring."

_Atmospheric Measurement Techniques, 2018_

## Referee Comment (RC1) · Anonymous Referee #1 · 28 Dec 2018

Review of Hanlon and Risk's 'Using computational fluid dynamics and field experiments to improve vehicle-based wind measurements for environmental monitoring'

Summary: The experiment, as suggested by the abstract, is well motivated and designed. The abstract is exceptionally well written. However, the paper does not meet the expectations from the abstract. I believe that major changes can make this work an extremely useful contribution.

Critical details regarding calculations are missing. The CFD work appears to be well

done, except for one very worrisome statement that I suspect the authors can easily explain. However, the CFD work is not useful as presented because it is not put in the context of a moving car, as could be done if the above mentioned equations were added to the paper. The bias adjustments presented in the field study appear to be an artifact of experimental design, rather than physically meaningful. This is again because the above mentioned equations were not used. The promise of meaningful use of the stationary anemometers was not fulfilled. These could have been used to make a correct assessment of biases as a function of the speed of the car (using the above mentioned equations), but this work was not done. While the authors are to be commended on obtaining useful observations and CFD calculations, they should redo the non-CFD work and present the final results in a context that combines the conclusions from the CFD results and field results. Such a reanalysis would make an extremely useful contribution. Furthermore, the field data are not made publicly available, which is preferred and often a requirement for modern journal publications.

Major Comments: 1) Section 2.1, end of first paragraph. Please say why these two sets of experiments appropriate to address the research objectives. I agree that they are appropriate, but some readers will not understand how they can be combined. This point is not addressed in the conclusions either, making this work needlessly incomplete. The relevant equations are presented in a variety of ways in

Smith, R. S, M. A. Bourassa, and R. J. Sharp, 1999: Establishing more truth in true winds. J. Atmos. Oceanic Technol., 16, 939-952.

2) Page 4: lines 10 and 11: The meaning of 'The drag coefficient was determined when the change in CD was less than 0.001.' is not clear. Is this condition sufficient? Or is convergence slow enough that this condition leads to large errors. How was it determined that this condition is sufficient, as 0.001 is rather a large faction of the drag coefficient over a smooth surface? The following sentences are insufficient for such a test. Granted, this situation becomes clear later, but it should be clear when presented. 3) Separating the experimental design (for two experiments) from the results (for two

experiments) is irritating and needlessly confusing. After the experimental design for the CFD I would greatly prefer to see the results from the CFD. a. Why are limitations of the CFD method/results discussed in the section on field observations? b. If this lack of results from flow into the wind is true (as literally read), then why should I read further? This seems like it could be critical flaw. 4) Are the observational data made publicly available? If so where? If not, should this work be published? 5) Page 10, Equation 1 and following line: What are the units of wind direction and wind speed? There will be quite a difference for m/s vs mph, and for radians vs. degrees! 6) Figure 6: It appears that there are fluctuations in the ratio of anemometer wind speed to vehicle wind speed that is associated with the speed of the car. This is not surprising because the measured speed is equal to the magnitude of 'the wind velocity minus the car velocity.' Such a dependency is expected as an artifact of the situation, and not as a systematic correction in the manner that the authors suggest. Showing the math requested in comment (1) would help the authors organize their thoughts and experiment in a manner that conveys useful information. I had expected the authors to use differences from the stationary anemometers to illustrate these biases, but that work is not contained in this paper. 7) Page 12, figure 8: What is actually shown in Fig. 8b? The caption and text claim that the figure shows the corrections, which suggests that the corrections are nearly identical to the measured winds. It appears that what is shown is the corrected winds.

Minor Comments: 1) Line 3: 'Used to study Meteorology' is too broad. Be more specific about the scale and the type of vehicle (for example, ships are vehicles, but not included in this study). 2) Table 1: Improve the caption to explain the labels of the table rows and columns. 3) Page 6, line7 and line 10-12: How were these corrections made? Was the car motion subtracted from the wind, or was the adjustment done correctly following Smith et al. 1999)? The statement that the calculations were done is R does not tell us how they were done. Smith, R. S, M. A. Bourassa, and R. J. Sharp, 1999: Establishing more truth in true winds. J. Atmos. Oceanic Technol., 16, 939-952. How was the temporal averaging done? See the above paper for the importance of the

correct averaging while wind directions are changing relative to the vehicle. 4) Page 7, line 3: 'similar qualitative trends' is so vague as to be nearly meaningless. 5) Page 7, line 10: This statement makes sense only for a fixed wind speed and direction. If it is being applied in general as suggested by the writing, then this result cannot be correct – or important caveats are missing. 6) Page 7, lines 21-22 & page 8, line 3 The logic '(where measured windspeed = true windspeed)' must be missing key caveats. 7) Page 7, line 34: 'How big is 'significant' in terms of heights? The existing text is too vague. 8) What is meant by wind speed > 3 standard deviations? Standard deviations of what (relatively to what)?  9) Page 9, line 2: What is meant by 'normalized wind speed'? 10) Page 10: At or near the end of the CFD results, please remind the reader that these are CFD results rather than field results. Changing the order of the paper to put method and results together would reduce this problem. a. The last line of this section belongs in the section on field results b. How was this yaw determimed? c. How was temporal averaging applied? Was it after individual calculations were corrected, or was the averaged corrected? The first approach is correct when the vehicle relative wind direction is changing.
* * *

---

## Referee Comment (RC2) · Anonymous Referee #2 · 29 Jan 2019

This study aims to guidance and correction factors for those aiming to use vehicle-based instruments to measure wind speed and direction. As such, it is a useful contribution to the literature. The paper is well organized and well written, but suffers from some problems with methods and the interpretation and presentation of results. It requires major revisions before being considered for publication.

General Comments

The reasons for focusing on installation of an instrument on top of a pickup cap are not

provided and not clear. Much of the initial discussion focuses on the work of Straka et al. (1996) and others that chose to put the anemometer out front of the vehicle to avoid the vehicle's flow field. And the authors show in their results (e.g. Fig. 3) that such a location would indeed be preferable. The authors need to be much more clear about the reasons for choosing to focus only on a location on top of a cap.

Problems with wind direction and speed data from a mobile instrument occur when the vehicle is experiencing acceleration (either changes in speed or direction). Data under such conditions should be removed from the analysis. However, this issue is not mentioned by the authors, even though it has a significant influence on both methods and results. I can only assume the authors have left these data in, and it helps to explain some of the large scatter in Fig. 7. This issue needs to be fully addressed.

Mobile wind data are collected at a 1 Hz interval, and fixed wind data are collected in one minute intervals (it is not clear if these are 1-min averages or not). The authors do not address challenges with comparing one data set to the other. Particularly in Fig. 8, it is hard to see how the mobile measurements and wind rose plots are an 'apples to apples' comparison. The authors need to address this issue.

The authors provide corrected wind data in Figs. 7 and 8 but readers (including myself) will want to see the uncorrected data in these plots as well. This will have the side benefit of making the plots larger and more legible.

Lastly, there is quite a bit of material relating to ships in the paper, and the reason is not entirely clear. Unless the authors can justify the inclusion of all of this material, it would be good to pare this down to essentials.

Detailed Comments

Page 2 Line 3 – There have been a number of other field studies that have made use of mobile measurements, including studies in Canada related to severe weather-related mesoscale meteorology (Taylor et al. 2011, Curry et al. 2017), air quality (Brook et al

2013) and urban meteorology (Joe et al. 2018). The authors should consider including these in the literature review and possibly make use of some of the results.

P4 L10 – re 22.222 m/s – do the authors believe the inlet velocity could be controlled to this accuracy? Please use a reasonable number of significant digits.

P4 L26 - To eliminate confusion here, it should read "The flow around a pickup truck with an open box is more complex than...because of wake interactions".

P4 L31 – Why is this the 'area of interest', and what evidence is there to support that flow atop the cap of the vehicle is located away from large pressure gradients?

P5 L5 - Were the speeds ranging from 40 to 100 also applied to these eight domains? Need to be more clear about this.

P5 L16 – Given known problems with measuring during acceleration, why make fixed measurements in the corners rather than the straight-aways? Why were the corners chosen in the first place?

P5 L19 – The meaning of this sentence is not clear – what is the 'frontal wind' from the vehicle? Please revise.

P6 L1 – Data should only be used when the vehicle speeds were kept constant – the authors do not mention this, and should fully explain their decisions here.

P6 L12 – I would like to see the detailed calculations included in an appendix.

P7 L25 – The meaning of this sentence is not clear to me. Please reword.

P8 L14 – It is not clear what these sentences are referring to – I see nothing in Table 2 that "shows" this.

Figure 7 – A few problems here – the grey bars need to be explained, another panel that shows the uncorrected mobile measurements needs to be included, and the separation of data at 0/360 degrees needs to be addressed so that there are only four 'bars' of

data, as in the control.

Page 11 L5 – 'data was' should be 'data were'

Figure 8 – A few issues here – the wind rose plot details are illegible particularly the frequency values (which appear to be missing entirely), a diagram the uncorrected mobile winds needs to be included, and the reason needs to be given to explain the WNW winds measured only along the top, headwind leg. The authors also need to specify what samples are being plotted here – certainly not 1 Hz data.

P12 L7 – The use of 'levels' as a verb here is confusing. Please reword.

P13 L4 – Over what periods are the data for the wind roses taken? This needs to be specified.

P13 L5 – Why not also average the winds over the leg and compare to the wind rose over that leg?

P13 L10 – The authors drove the route at speeds of 40, 50, 60, 70, and 80 km/h but only present results from 70 km/h in Fig 8. They need to also show results from the other speeds (perhaps best using averages).

P13 L10 – Re "improved", it is difficult to see this and is not quantative. The authors need to better support/interpret the results.

P13 L15 – I understand that CFD cannot simulate this but it could use a clearer explanation.

P13 L26 – Why is there more variability in tailwind conditions? Please explain for readers.

P13 L30 – Perhaps this could be better explain – the tail wind will be 'embedded' in the flow around the vehicle in real-world conditions, so why wouldn't it be 'detected'?

P14 L1 – Should be 'mobile anemometers'

P14 L1 – Not sure where the 'mean' was compared. Please expand on this.

P14 L7 – What about other uses of mobile wind data? How would this improve a meteorological field study, for example?

P15 L20 – On a daily basis? Only during field studies?

P15 L30 – But what is the representative height being aimed for? You could try to install at 20 m mast and it would certainly be out of the vehicle envelope, but do you want to know the winds at that height? The authors have not made it clear at what height stakeholders require wind data.

References

Brook, J. R., P. A. Makar, D. M. L. Sills, K. L. Hayden and R. McLaren, 2013: Exploring the nature of air quality over southwestern Ontario: main findings from the Border Air Quality and Meteorology Study. Atmos. Chem. Phys., 13, 10461–10482, 10.5194/acp-13-10461-2013.

Curry, M., J. Hanesiak, S. Kehler, D. M. L. Sills, and N. Taylor, 2017: Ground-based observations of the thermodynamic and kinematic properties of lake-breeze fronts in southern Manitoba, Canada. Boundary-Layer Meteorol., 163, 143–159, 10.1007/s10546-016-0214-1.

Joe, P., S. Bélair, N. B. Bernier, V. Bouchet, J. R. Brook, D. Brunet, W. Burrows, J. P. Charland, A. Dehghan, N. Driedger, C. Duhaime, G. Evans, A.-B. Filion, R. Frenette, J. de Grandpré, I. Gultepe, D. Henderson, A. Herdt, N. Hilker, L. Huang, E. Hung, G. Isaac, C.-H. Jeong, D. Johnston, J. Klaassen, S. Leroyer, H. Lin, M. MacDonald, J. MacPhee, Z. Mariani, T. Munoz, J. Reid, A. Robichaud, Y. Rochon, K. Shairsingh, D. Sills, L. Spacek, C. Stroud, Y. Su, N. Taylor, J. Vanos, J. Voogt, J. M. Wang, T. Wiechers, S. Wren, H. Yang, T. Yip, 2018: The Environment Canada Pan and ParaPan American Science Showcase Project. Bull. Amer. Meteorol. Soc., 921-953, DOI 10.1175/BAMS-D-16-0162.1

Taylor, N. M., D. M. L. Sills, J. M. Hanesiak, J. A. Milbrandt, C. D. Smith, G. S. Strong, S. H. Skone, P. J. McCarthy, and J. C. Brimelow, 2011: The Understanding Severe Thunderstorms and Alberta Boundary Layers Experiment (UNSTABLE) 2008. Bull. Amer. Meteorol. Soc., 92, 739-763.
* * *

---

## Author Comment (AC1) · 25 Apr 2019

We would like to thank Anonymous Reviewer #2 for their time in suggesting changes that will improve our manuscript. Our responses to the review comments are provided below in blue.

General Comments:

The reasons for focusing on installation of an instrument atop of a pickup cap are not provided and not clear. Much of the initial discussion focuses on the work of Straka et

al. (1996) and others that chose to put the anemometer out front of the vehicle to avoid the vehicle's flow field. And the authors show in their results (e.g. Fig. 3) that such a location would indeed be preferable. The authors need to be much more clear about the reasons to choosing to focus only on one location on top of a cap.

Problems with wind direction and speed data from a mobile instrument occur when the vehicle is experiencing acceleration (either changes in speed or direction). Data under such conditions should be removed from the analysis. However this issue is not mentioned by the authors, even though it has a significant influence on both methods and results. I can only assume the authors have left these data in, and it helps to explain some of the large scatter in Fig. 7. This issue needs to be fully addressed.

The authors provide corrected wind data in Figs. 7 and 8 but readers (including myself) will want to see uncorrected data in these plots as well. This will have the side benefit of making the plots larger and more legible.

Lastly, there is quite a bit of material relating to ships in the paper, and the reason is not entirely clear. Unless the authors can justify the inclusion of all of this material, it would be good to pare this down to essentials.

We appreciate these general comments. We have used these general remarks to direct each detailed change suggested in comments below.

Detailed Comments

1. Page 2 Line 3 – There have been a number of other field studies that have made use of mobile measurements, including studies in Canada related to severe weather-related mesoscale meteorology (Taylor et al. 2011, Curry et al. 2017), air quality (Brook et al. 2013) and urban meteorology (Joe et al. 2018). The authors should consider including these in the literature review and possibly make use of some of the results.

We have considered the recommended literature, and updated the manuscript with the following:

A reference to Curry et al., 2017 and Taylor et al., 2011 has been added to Page 2 Line 3 of the manuscript.

We have used the literature to add to the discussion in the manuscript. We comment on the updates in detailed comments 25 and 26.

2. P4 L10 – re 22.222 m/s – do the authors believe the inlet velocity could be controlled to this accuracy? Please use a reasonable number of significant digits.

The text has been modified, 22.222 m/s has been replaced with 22.2 m/s.

3. P4 L26 - To eliminate confusion here, it should read "The flow around a pickup truck with an open box is more complex than...because of wake interactions".

P4 L26 has been modified to "The flow field around a pickup truck is more complex than the flow around an SUV of sedan because of wake flow interactions."

4. P4 L31 – Why is this the 'area of interest', and what evidence is there to support that flow atop the cap of the vehicle is located away from large pressure gradients?

The text "Our area of interest, the flow atop the cap of the vehicle is located away from large pressure gradients. Because the cap eliminates large pressure gradients at the area of interest, k-$\varepsilon$ performance in the drag comparison, and its results in other studies, we have selected the k-$\varepsilon$ turbulence model as appropriate for our analysis in the velocity field above a capped pickup truck."

Has been replaced with,

"Our motivation for placing an anemometer atop a pick-up truck came to assist vehicle-based gas monitoring systems (Atherton et al., 2017; Baillie et al., 2019; O'Connell et al., 2019) measuring gas emissions to achieve practical placement and further calibration practices. In these studies, the anemometer was mounted above a truck cap. Because of the differing vehicle shape from an open bed pickup truck, k-$\varepsilon$ performance in the drag comparison, and performance in other studies, we have selected the k-$\varepsilon$

turbulence model as appropriate for our analysis in the velocity field above a capped pickup truck."

We removed the statement , "Because the cap eliminates large pressure gradients at the area of interest" in the methods portion of the manuscript, as we have not yet presented our CFD results.

5. P5 L5 - Were the speeds ranging from 40 to 100 also applied to these eight domains? Need to be more clear about this.

Yes, the same speed ranges were applied to all eight domains. The following text has been added to P5 L10:

"We used the additional computational domains to test all combinations of yaw under inlet speeds ranging from 40 km h-1 to 100 km h-1 in 5 km h-1 increments.

6. P5 L16 – Given known problems with measuring during acceleration, why make fixed measurements in the corners rather than the straight-aways? Why were the corners chosen in the first place?

The following text has been added to P5 L17:

"The stationary anemometers were placed on the corners to provide opportunity for comparison with both legs of the straightaway. This ensured that we would have data for comparison with each leg should an anemometer malfunction.

We consistently found locations with flat, treeless terrain and no surrounding infrastructure at locations approximately 300 m from each corner. The accessibility of these stationary anemometer sites also influenced our selection of placing the stationary anemometers near the corners.

7. P5 L19 – The meaning of this sentence is not clear – what is the 'frontal wind' from the vehicle? Please revise.

This sentence was unclear and not necessary, it has been removed.

8. P6 L1 – Data should only be used when the vehicle speeds were kept constant – the authors do not mention this, and should fully explain their decisions here.

This is an excellent point. The objective of our field study was to collect measurements at fixed vehicle speeds that could be compared with stationary anemometers. The measurements from the corners when the vehicle was turning, and accelerating up to cruise control speed should be excluded. The following text has been inserted to P6 L6:

"The 1Hz geolocation measurements were used to compute the vehicle speed and bearing, field measurements that deviated more than 2.5 km h-1 from the implemented cruise control speed, vehicle bearing measurements that differed more than 5° from designated vehicle course were removed from the data set."

Figure 7 and 8 have been modified to exclude non-steady-state measurements.

9. P6 L12 – I would like to see the detailed calculations included in an appendix.

We removed the text, "and a true wind vector was computed by removing the vehicle vector from the vehicle wind vector. All computations were completed using R 3.4.1 statistical software (R Core Team. 2016)"

And replaced it with, "We calculated the vehicle's course over the ground using the GPS coordinates to obtain a vector of equal magnitude and opposite direction to be the frontal wind induced by the vehicle's motion (F). We compute the meteorological wind vector (W) by subtracting the frontal wind induced by the vehicle's motion (F) from the raw anemometer measured wind vector (A). The calculation is presented in Equation (1) . The computation comes from the method calculating the meteorological true wind from a moving vessel presented by Smith et al., 2009.

$W = A - F$

We have included a figure and details describing the vector translation required in the computation to S1 in the supplementary information.

10. P7 L25 – The meaning of this sentence is not clear to me. Please reword.

P7 L25 has been reworded to say "The second set of simulations in this study are of particular interest because they evaluate the flow field under cross wind conditions. Many previous CFD and wind tunnel studies (Yang and Khalighi 2009; Holloway et al., 2009) examine the flow field resulting from inlets, which direct the airflow along the longitudinal axis of the vehicle. Aside from Houston et al. 2016, we have not found studies using CFD to conduct crosswind experiments to study the flow field around a vehicle.

11. P8 L14 – It is not clear what these sentences are referring to – I see nothing in Table 2 that "shows" this.

The two sentences beginning on P8 L14 have been reworded to say:

"Table 2 shows that the height required for an anemometer to experience bias below 2% increases with increasing yaw angle. Anemometer heights selected through frontal wind tunnel tests would still experience bias under cross wind conditions."

12. Figure 7 – A few problems here – the grey bars need to be explained, another panel that shows the uncorrected mobile measurements needs to be included, and the separation of data at 0/360 degrees needs to be addressed so that there are only four 'bars' of C3 data, as in the control.

The following changes have been made to figure 7 to address both the above detailed comment and the general comments made by the reviewer.

The measurements have been regrouped so that the figure shows measurements corresponding to vehicle bearing measurements of 0°, 90°, 180°, 270° corresponding to the vehicle travelling in the North, East, South and West directions. This change removes the separation of the data at 0° and 360°.

The grey bars represent measurements that were collected in tail wind conditions. A second sentence has been added to the figure caption that reads, "The shaded

grey bars represent measurements acquired when the vehicle was driving in tail wind conditions."

The change addressed in detailed comment 8 reduces the scatter in this figure. Figure 7 now includes measurements where the vehicle speed deviates no more than 2.5 km h-1 from the implemented cruise control speed, and the vehicle bearing measurements differ no more than 5° from designated vehicle course.

13. Page 11 L5 – 'data was' should be 'data were'

This change has been made

14. Figure 8 – A few issues here – the wind rose plot details are illegible particularly the frequency values (which appear to be missing entirely), a diagram the uncorrected mobile winds needs to be included, and the reason needs to be given to explain the WNW winds measured only along the top, headwind leg. The authors also need to specify what samples are being plotted here – certainly not 1 Hz data.

The following changes have been made to figure 8:

The measurements presented in the figure have been modified to reflect the change made in response to detailed comment 8, removing measurements outside of the speed test. The data was previously the 70 km h-1 speed test, but was replaced with the 60 km h-1 speed test because fewer measurements were excluded as a result of being 2.5 km h-1 outside the speed test range.

The wind rose frequency values are now legible. The stationary anemometers report one minute averages. The instrument collects a wind speed and direction measurement every 10 seconds and reports the average of those six. The instrument also reports a wind gust measurement which is the highest instantaneous wind speed measured during the selected averaging interval. The wind roses are the average wind direction measurement and wind gust reported for each minute. The wind roses contain 14 minutes of data, and are the measurements acquired during the 60 km h-1

speed test.

Adding another plot with raw measurements to Figure 8 is not ideal for space, this plot is added to appendix A.

15. P12 L7 – The use of 'levels' as a verb here is confusing. Please reword.

P12 L7 has been reworded to say, "Applying the empirical correction reduces the difference in the mean of the wind direction measurements across the four vehicle bearings."

16. P13 L4 – Over what periods are the data for the wind roses taken? This needs to be specified.

The data displayed in the wind roses was taken over a 14 minute period, which was the duration of the 60 km h-1 field test. This information has been added to the manuscript to further describe 8.

The following text has been added to P13 L4, " The measurements in the wind roses are the wind gust and average wind direction measurements reported each minute of the 14 minute time speed test."

17. P13 L5 – Why not also average the winds over the leg and compare to the wind rose over that leg?

Figure 8 has been modified to include a table comparing the mobile measurements averaged over the leg with the average from the wind rose.

The wind rose at each corner is a representation of the wind patterns we would expect to see across the survey leg. We did not obtain raw one second measurements from the stationary anemometers making it difficult to average the measurements in an identical way. Our study instead uses the stationary measurements as a guide to evaluate the accuracy and precision of the 1 Hz vehicle-based anemometer measurements.

18. P13 L10 – The authors drove the route at speeds of 40, 50, 60, 70, and 80 km/h but only present results from 70 km/h in Fig 8. They need to also show results from the

other speeds (perhaps best using averages).

Figure 8 was constructed to show – in detail - the variability in the mobile wind measurements after the CFD calibration was applied. It was more the purpose of Figure 7 to show the general impact of the CFD calibration across all speed tests.

19. P13 L10 – Re "improved", it is difficult to see this and is not quantative. The authors need to better support/interpret the results.

The text "Figures 7 and 8 demonstrate that measurement reliability and likely accuracy is improved by applying an empirical correction to vehicle anemometer measurements prior to correcting for the vehicle vector. The empirical correction from CFD improved the field measurements, but still could improve in tail wind."

Has been changed to "Figure 8 shows that applying the CFD calibration improves the agreement of the average mobile wind speed and direction measurements with the average stationary measurements in three of the four legs of the route."

The text "In the top plot, black arrows representing wind speeds over 50 km h-1 are present. In the bottom plot, these arrows are replaced with wind speeds 6-8 km h-1 lower. The empirically-corrected data appears to match the stationary measurements better in tail wind, but still experiences difficulty with direction." has been removed.

20. P13 L15 – I understand that CFD cannot simulate this but it could use a clearer explanation.

The text has been replaced with: "Field measurements were the only method of obtaining data under tail wind conditions in our study. Our CFD model was limited to a single inlet, and did not have the capability to simultaneously model oncoming air from the vehicle's motion, and wind blowing from behind the truck."

21. P13 L26 – Why is there more variability in tailwind conditions? Please explain for readers.

This comment is addressed in the next detailed comment response.

22. P13 L30 – Perhaps this could be better explain – the tail wind will be 'embedded' in the flow around the vehicle in real-world conditions, so why wouldn't it be 'detected'?

We have added the bolded text in at P13 L30 to address detailed comments 21 and 22 together, into the paragraph beginning at P13 L24. The text has been changed to:

"The plotted wind vectors in Fig. 8 show that the 1 Hz mobile wind measurements experience the most variability in speed and direction when the vehicle is travelling in tail wind conditions. The empirically corrected tail wind direction measurements were in reasonable (within $\sim 30°$) agreement with stationary direction measurements on field days with winds greater than 25 km h-1, but deviated on days with lower wind speeds. When the vehicle speed is much larger than the wind speed, the vehicle creates a continuous flow field that can obstruct natural winds. The anemometer detects continuous frontal wind from the vehicle, and becomes less sensitive to components of the wind coming from behind the vehicle. It is likely that a windspeed threshold for the magnitude of tail wind detected by the mobile anemometer, and that this threshold varies with vehicle speed. Additional field testing on field days with low winds is recommended to evaluate the quality of tail wind measurements, identify windspeed thresholds, and develop quality control criteria for tail wind measurements. As CFD is limited in developing an empirical correction for tail wind, additional filed data is recommended to develop a correction for tail wind measurements.

23. P14 L1 – Should be 'mobile anemometers'

This change has been made.

24. P14 L1 – Not sure where the 'mean' was compared. Please expand on this.

We have moved the text, "The results are displayed in Table 3." From P14 L4 to P14 L2 to draw attention to Table 3 which compares the percent difference between mean short and tall anemometer measurements across vehicle speed.

25. P14 L7 – What about other uses of mobile wind data? How would this improve a meteorological field study, for example?

In the discussion we have added the statement, "Vehicles outfitted to study severe weather (Curry et al., 2017; Taylor et al., 2011), and urban meteorology (Joe et al., 2018) show resilience to bias by also outfitting vehicles with wind sensors above and ahead of the vehicle."

This comment is also addressed in the response to the detailed comment below.

26. P15 L20 – On a daily basis? Only during field studies?

The following text has been added to P15 L24

"Vehicle-based wind measurements from field studies can be used to contribute to detailed observing networks of specific sites. Furthermore, vehicle-based wind measurements can be collected conveniently by vessels of opportunity that travel routine routes, and contribute to weather data assimilation to evaluate accuracy for air quality and weather forecasting models (Brook et al., 2013).

27. P15 L30 – But what is the representative height being aimed for? You could try to install at 20 m mast and it would certainly be out of the vehicle envelope, but do you want to know the winds at that height? The authors have not made it clear at what height stakeholders require wind data.

P15 L32 has been modified to:

"For applications requiring near surface wind measurements to be paired with other vehicle-based measurements, and similar anemometer placements to those in this study, we recommend anemometers on truck caps be mounted at least 1 m above the vehicle, and that an empirical correction be applied. While mounting the anemometer at larger heights above the vehicle reduces the impact of flow distortion resultant from the vehicle's motion, placement must be practical and safe for road travel. For larger vehicle's, we expect mounting anemometers on vehicle's at heights much greater than

1 m to be potentially impractical. It is important that placement keeps the sensor below common road clearance limits for bridges and underpasses."

---

## Author Comment (AC2) · 25 Apr 2019

We would like to thank Anonymous Reviewer #1 for their time in suggesting changes that will improve our manuscript. Our responses to the review comments are provided below.

Summary: The experiment, as suggested by the abstract, is well motivated and designed. The abstract is exceptionally well written. However, the paper does not meet the expectations from the abstract. I believe that major changes can make this work

an extremely useful contribution. Critical details regarding calculations are missing. The CFD work appears to be well done, except for one very worrisome statement that I suspect the authors can easily explain. However, the CFD work is not useful as presented because it is not put in the context of a moving car, as could be done if the above mentioned equations were added to the paper. The bias adjustments presented in the field study appear to be an artifact of experimental design, rather than physically meaningful. This is again because the above mentioned equations were not used. The promise of meaningful use of the stationary anemometers was not fulfilled. These could have been used to make a correct assessment of biases as a function of the speed of the car (using the above mentioned equations), but this work was not done. While the authors are to be commended on obtaining useful observations and CFD calculations, they should redo the non-CFD work and present the final results in a context that combines the conclusions from the CFD results and field results. Such a reanalysis would make an extremely useful contribution. Furthermore, the field data are not made publicly available, which is preferred and often a requirement for modern journal publications.

We appreciate this summary and have used the comments to make changes as suggested in the detailed comments, where the reviewer expands on these general comments.

Major Comments:

1) Section 2.1, end of first paragraph. Please say why these two sets of experiments of experiments appropriate to address the research objectives. I agree that they are appropriate, but some readers will not understand how they can be combined. This point is not addressed in the conclusions either, making this work needlessly incomplete. The relevant equations are presented in a variety of ways in:

Smith, R. S, M. A. Bourassa, and R. J. Sharp, 1999: Establishing more truth in true winds. J. Atmos. Oceanic Technol., 16, 939-952.

[Figure]

We have added the following text at to P3 L9 in section 2.1:

"By evaluating the vehicle's external flow field under both varying vehicle speed and wind yaw angle we can quantify the bias the vehicle shape induces on a truck mounted anemometer and calibrate the measurements prior to correcting wind speed and direction measurements for the vehicle's motion."

The following text has been added into the Results and Discussion section on P10 L6:

"The empirical calibration must be applied to remove the bias of the vehicle's shape on anemometer measurements prior to correcting for the vehicle motion."

The above reference has also been used to address minor comment 3 and add further description to the field methods.

Page 4: lines 10 and 11: The meaning of 'The drag coefficient was determined when the change in CD was less than 0.001.' is not clear. Is this condition sufficient? Or is convergence slow enough that this condition leads to large errors. How was it determined that this condition is sufficient, as 0.001 is rather a large faction of the drag coefficient over a smooth surface? The following sentences are insufficient for such a test. Granted, this situation becomes clear later, but it should be clear when presented.

We wanted the CFD results to match the instrument accuracy available in our field tests. As our model was simplified, we knew we could not expect perfect accuracy in the drag coefficient. 0.001 is less than 1% of the manufacturers reported drag coefficient (0.386), and less than the instrument errors used in the collection of field measurements.

3) Separating the experimental design (for two experiments) from the results (for two experiments) is irritating and needlessly confusing. After the experimental design for the CFD I would greatly prefer to see the results from the CFD. a. Why are limitations of the CFD method/results discussed in the section on field observations? b. If this lack of results from flow into the wind is true (as literally read), then why should I read

further? This seems like it could be critical flaw.

We have restructured the manuscript to address this point. We have inserted a discussion section at P14 L7. In this section we discuss the results together, and the limitations from CFD are discussed in this section and no longer in the field results.

4) Are the observational data made publicly available? If so where? If not, should this work be published?

Field data have been made available at: https://figshare.com/articles/Mobile_and_Stationary_Anemometer_Wind-SoutheasternSK_May2017_csv/8035235",

doi = "10.6084/m9.figshare.8035235.v1"

5) Page 10, Equation 1 and following line: What are the units of wind direction and wind speed? There will be quite a difference for m/s vs mph, and for radians vs. degrees!

We have modified presentation of the equation to be more descriptive with assigned variables and include units. The equation was modified to include both the anemometer windspeed (AWS,) and the corrected windspeed (WS) for clarity.

The following text has been added to P10 L1 :

"The polynomial pictured in Fig 5 is multiplied by the Anemometer windspeed (AWS), to give the corrected wind speed (WS). Equation 2 gives the side-mounted anemometer's correction function for wind direction (WD) measurements ranging from -40° < WD < 40°. Wind speed units are in km h-1 and wind direction units are in degrees.

6) Figure 6: It appears that there are fluctuations in the ratio of anemometer wind speed to vehicle wind speed that is associated with the speed of the car. This is not surprising because the measured speed is equal to the magnitude of 'the wind velocity minus the car velocity.' Such a dependency is expected as an artifact of the situation, and not as a systematic correction in the manner that the authors suggest. Showing the math requested in comment (1) would help the authors organize their thoughts and

experiment in a manner that conveys useful information. I had expected the authors to use differences from the stationary anemometers to illustrate these biases, but that work is not contained in this paper.

The objective of this figure is to show that the short anemometer placement consistently measures higher wind speeds than the tall anemometer placement. The manuscript has been edited to better describe the figure and its importance of anemometer height a top the vehicle.

The statement , "Figure 6 shows the tall and short anemometer measurements scaled with vehicle speed" has been modified to say "Figure 6 shows magnitude of the tall and short anemometer measurements when normalized by the vehicle speed."

The reference to Smith et al., 2009 from comment 1 has been used to add an equation and text to P7 L4. Please refer to our response to minor comment 3 and detailed comment 9 in our Response to RC2 for the detailed changes.

7) Page 12, figure 8: What is actually shown in Fig. 8b? The caption and text claim that the figure shows the corrections, which suggests that the corrections are nearly identical to the measured winds. It appears that what is shown is the corrected winds.

Please refer to our Response to RC2 for revisions proposed for Fig. 8.

Minor Comments: 1) Line 3: 'Used to study Meteorology' is too broad. Be more specific about the scale and the type of vehicle (for example, ships are vehicles, but not included in this study).

The text has been replaced to say, "Currently, land vehicle-based wind measurements are used to study severe weather-related mesoscale meteorology (Belusic et al., 2014, Straka et al., 1996), lake meteorology (Taylor et al. 2011; Curry et al. 2017) and are integrated into methane measurement studies to detect, quantify, and map emission plumes from oil and gas developments (Atherton et al., 2017; et al., 2015; Zazzeri et al., 2015).

2) Table 1: Improve the caption to explain the labels of the table rows and columns.

The caption has been changed to "Drag comparison of three turbulence models with inlet velocity of 22.2 m/s. The drag coefficient (CD) for each turbulence model is reported with the number of iterations required to achieve a solution and the percent difference ($\Delta$CD) from the manufacturers reported CD of the vehicle model."

The turbulence models in the rows of the table are described on P4 L7 before introducing the table.

3) Page 6, line7 and line 10-12: How were these corrections made? Was the car motion subtracted from the wind, or was the adjustment done correctly following Smith et al. 1999)? The statement that the calculations were done is R does not tell us how they were done. Smith, R. S, M. A. Bourassa, and R. J. Sharp, 1999: Establishing more truth in true winds. J. Atmos. Oceanic Technol., 16, 939-952. How was the temporal averaging done? See the above paper for the importance of the correct averaging while wind directions are changing relative to the vehicle.

We have added the following text to P6 L7:

"To correctly compute true winds, vehicle based wind must be corrected for the vehicle's motion over the fixed earth. The vehicle vector has a direction equivalent to the vehicle's course over the ground, and a magnitude equivalent to the vehicle's speed over the ground (Smith et al., 1999)."

Please also refer to detailed comment #9 our Response to RC2, where we describe the subtraction equation.

The averaged vehicle-based anemometer measurements presented in Figure 8 were calculated by expressing all of the wind speed and direction measurements in complex exponential form to keep wind speed and direction as one vector, as opposed to separating them into components. Figure S1 in the supplement details how the components of each vector were calculated, and how the averaging was done.

4) Page 7, line 3: 'similar qualitative trends' is so vague as to be nearly meaningless.

P7 L3 has been modified to say, "The CFD and field experiments showed that the measured wind at the anemometer location varied under wind yaw angle. The bias found in the CFD varied with the rotation of the truck, and the field results concluded that the measured wind differed in head, cross and tail wind conditions."

5) Page 7, line 10: This statement makes sense only for a fixed wind speed and direction. If it is being applied in general as suggested by the writing, then this result cannot be correct – or important caveats are missing.

P7 L10 has been modified to:

"In each computational domain, we found that the wind bias scaled with vehicle speed and the amount of bias (slope) differed with the location above the vehicle."

6) Page 7, lines 21-22 & page 8, line 3 The logic '(where measured windspeed = true windspeed)' must be missing key caveats.

The following text on P7 L21: " This observation is similar to a conclusion by Moat et al. (2005), stating that shipboard anemometers should not be placed close to the line of equality (where measured wind speed = true wind speed) as high pressure gradients are present in this region. The line of equality would be expected to move vertically to some degree, according to the ground and/or wind speed."

References the work Moat et al., conducted on airflow distortion at anemometer sites on ships. As part of the study, the authors used Vectis CFD code to model the flow over generic voluntary observing ship models. The study presents the flow pattern above a bulk carrier model, and shows that above the bridge there exists a decelerated region of flow and at a greater height above the bridge, an accelerated region of flow. Between the decelerated region and accelerated region of flow there exists a line where the flow is equivalent to the wind speed. In this region the velocity gradients are very steep, and sensitive to bias. Moat et al., concluded that an anemometer should not be placed in

this region.

In our CFD model, we also found that a decelerated region of flow existed below the accelerated region of flow. We recommend placing the anemometer above this region to avoid the steep velocity gradients, and referenced this text to support our results.

7) Page 7, line 34: 'How big is 'significant' in terms of heights? The existing text is too vague.

P 7 L 34 has been replaced with "When exposed to large yaw angles, the wind bias over the truck at low heights can be twenty percent, and even at a height of 1.7 m above the vehicle, a bias of greater than 5 percent is present."

8) What is meant by wind speed > 3 standard deviations? Standard deviations of what (relatively to what)?

Good point. This information was lacking in the manuscript. The yellow circles represent wind speed measurements that are greater than 3 standard deviations away from the mean wind speed measurement. The mean wind speed measurement is calculated specific to a) the wind speed measurements from the vehicle-based anemometer after the frontal-correction was applied, b) the wind speed measurements from the vehicle-based anemometer after the empirical correction, and then frontal correction was applied, c) the wind speed measurements from the stationary anemometer located at the 4 different vehicle bearings.

The caption of Figure 7 has been modified to include: " The yellow circles represent wind speed measurements that are greater than three standard deviations away from the mean wind speed of the wind speed measurements calculated with the a) frontal correction applied, b) empirical correction and frontal correction applied, and c) specific to the stationary anemometer at each bearing."

9) Page 9, line 2: What is meant by 'normalized wind speed'?

The normalized wind speed is the measured wind speed divided by the inlet velocity.

The text has been changed to:

"The normalized wind speed (measured wind speed: inlet speed) at the location of an anemometer mounted in the same location as the field test was computed for each yaw angle simulation."

10) Page 10: At or near the end of the CFD results, please remind the reader that these are CFD results rather than field results. Changing the order of the paper to put method and results together would reduce this problem. a. The last line of this section belongs in the section on field results b. How was this yaw determimed? c. How was temporal averaging applied? Was it after individual calculations were corrected, or was the averaged corrected? The first approach is correct when the vehicle relative wind direction is changing.

a) This has been addressed with by the addition of a discussion section in response to major comment 3. b) The yaw is determined by the raw anemometer wind direction measurements. c) There is no temporal averaging on the mobile measurements.

Please also note the supplement to this comment:
https://www.atmos-meas-tech-discuss.net/amt-2018-354/amt-2018-354-AC2-supplement.pdf

---

## Author Comment (AC3) · 25 Apr 2019

**Supplemental Material**

**Using computational fluid dynamics and field experiments to improve vehicle-based wind measurements for environmental monitoring.**

*Tara Hanlon [1]\*, David Risk [1],*

[1] *St. Francis Xavier University, Department of Earth Sciences, Antigonish Nova Scotia, Canada*

Corresponding Author: Tara Hanlon
Email: thanlon@stfx.ca

**Table of Contents**

[Figure]

Figure S1: Illustration of using complex exponential form to keep the wind speed and wind direction as one vector, as opposed to separating them into components.

**S1. Method of Vector Translation**

Measured wind vectors relative to the vehicle must be translated to true wind vectors relative to the ground. This correction requires working in three different reference frames: the truck frame, the math frame (cartesian or complex), and the geographic frame. We had to first calculate the vehicle vector from the GPS coordinates in the geographic coordinate system, then translate the anemometer wind vector from the truck's coordinate system using the GPS coordinates, before

lastly removing the vehicle vector from the anemometer wind vector. We expressed all vectors in terms of the complex plane. In Figure S1, we expressed the wind vector in terms of the magnitude (WS) and angle ($\theta$) in complex exponential form (WS x $e^{i\theta}$). We also used trigonometric functions to display the real component of the vector WS x cos($\theta$) and the imaginary component of the vector WS x sin($\theta$). We expressed all vectors in complex exponential form instead of using trigonometric functions to calculate vector components. To average measurements, we averaged the real and imaginary components to create a resultant vector. The calculated averaged wind speed and wind direction are the magnitude and direction of the resultant vector.

**S2. *Raw Speed Measurements**

[Figure]

(b) Raw wind vector measurements
collected when within 2.5 km h$^{-1}$ of 60 km h$^{-1}$.

**Figure S2.** *Uncorrected anemometer measurements sampled during the 60 km h$^{-1}$ field test.*

Figure S2 shows raw wind measurements to complement figure 8.

---

## Author Response (AR2)

**Response to Review**

amt-2018-354
Report #1, Anonymous referee #1

Title: Using computational fluid dynamics and field experiments to improve vehicle-based wind measurements for environmental monitoring.

Authors: Tara Hanlon*, David Risk

Journal: Atmospheric Measurement Techniques

We would like to thank Anonymous Reviewer #1 for their time in suggesting changes that will improve our manuscript. Our responses to the review comments are provided below in blue.

Comments

Page 2 – re references for "severe weather- related meteorology", the authors need to include Taylor et al. 2011 since this reference is used later in the manuscript (page 16) for the same reason.

This change has been made.

Page 4 – re the last sentence, it is awkwardly worded – suggest "…motivation for placing an anemometer atop a pickup truck is to assist…measuring gas emissions in order to achieve…". Consider breaking into two sentences – that may help with clarity as well.

The text has been changed to "Our motivation for placing an anemometer atop a pickup truck is to assist vehicle-based gas monitoring systems (Atherton et al., 2017; Baillie et al., 2019; O'Connell et al., 2019) measuring gas emissions to achieve practical placement and further calibration practices."

Page 6 – Re "The tripods were located approximately 300 m clockwise from each corner." The authors still need to explain **why** this choice was made i.e. why 300 m from the corner. They just need to expand this sentence.

The text has been modified to "The tripods were located approximately 300m clockwise from each corner to allow for truck to be travelling at a constant speed when passing the stationary instrument."

Page 7 – re "to be the frontal wind", omit "be"

This change has been made.

Page 7 – I don't think there is a need to express that simple equation mathematically, the verbal description is fine.

This change was requested from another reviewer, we have decided to include it in the manuscript to provide clarity.

Page 8 – re "high pressure gradients", easily confused with "high pressure" suggest using large pressure gradients"

Thank you for your suggestion. This change has been made.

Page 9 – re "measured wind speed: inlet speed", it is easy to miss that this is a ratio – please fins another way to express this.

This text has been expressed as a fraction to indication the ratio. ( "$\frac{measured\ wind\ speed}{inlet\ speed}$" )

Page 10 – re "multiplied by the Anemometer wind speed (Aws) to give the ..", I don't think anemometer should be capitalized here (though I can see why it was), and the comma either needs to be removed or corrected.

The comma has been removed.

Page 12 – in the first sentence, using 'true' here is confusing, especially since it is used in two different senses. Suggest removing the first "true" - the meaning is not clear.

We believe this sentence being referenced is on page 10, the first true has been removed.

Page 12 – re "Figure 6 shows magnitude of .." should be the magnitude of"

This change has been made.

Page 13 – re "72 % of raw…" should be "72% of raw…"

This change has been made.

Figure 8 – I think the figure labels for (a) and (b) are confusing. The authors should place the (a) label at the top of the figure and the (b) label between the diagrams but leaving enough white space so that it is clear (b) relates to the bottom diagram.

The captions have been moved to above the diagram they are describing.

Page 15 – shoud be "cross-wind measurements" with the hyphen

This change has been made.

Page 16 – re "measurement methods attributed to …"should be "measurement methods contributed to…"

This change has been made.

Page 17 – should be "fixed-site measurements"

It is unclear what sentence on page 17 this comment refers to. If the reviewer can provide more detail we will revisit.

Page 17 – in conclusions, "CFD-derived corrections can help", this needs to be expanded i.e. can help to what?

The text has been expanded to, "CFD-derived corrections can help to reduce bias."

Page 17 – last paragraph, "resultant" should be "resulting" and vehicle's" should be "vehicles".

This change has been made.

Page 17 – the last sentence needs to be expanded upon. It's not enough to say that they are "appropriate" the authors need to summarize why they found they were appropriate. Perhaps this last sentence can be moved up with the 'can help' sentence mentioned above, while expanding on the meaning.

This sentence has been changed to,

"The calibration of wind speed and direction measurements through CFD-derived empirical calibrations was shown to be effective in reducing the bias that results from the vehicle's air flow field. Field measurements with good control data can also provide datasets for developing calibrations."

**Response to Review**

amt-2018-354-RC1, 2019

Title: Using computational fluid dynamics and field experiments to improve vehicle-based wind measurements for environmental monitoring.

Authors: Tara Hanlon*, David Risk

Journal: Atmospheric Measurement Techniques

We would like to thank Anonymous Reviewer #2 for their time in suggesting changes that will improve our manuscript. Our responses to the review comments are provided below in blue.

General comments:

Overall, this manuscript presents an interesting analysis of the biases that occur when trying to measure winds on a moving vehicle. As this is a common practice for a number of land-based research and operational activities (e.g., severe storms research, chemical plume monitoring), the analysis and discussions presented in the manuscript will contribute valuable information to future users of vehicle-based wind sensors.

My primary concerns with the manuscript are that in many places the writing is unclear or written in an overly-complicated (sometimes too condensed) manner, making the results hard to follow. The manuscript reads as a technical report for a very specialized set of domain area (e.g., CFD) experts who already understand all the terminology. This is not appropriate for a journal article and more needs to be done to expain/clarify the methods and results. Also, there were some missed opportunities to do some direct comparisons between the stationary and mobile wind data that would have strengthened the manuscript.

We appreciate these general comments and and have used the comments to make changes as suggested in the specific comments provided by the reviewer.

Specific Comments:

1) Abstract line 14-15 The authors note the difference between the stationary and mobile anemometers increased with yaw angle, but nowhere in the paper are these results shown. I would expect a table or a graphic showing a comparison between the stationary and

mobile anemometers that are coincident in time (as provided in your supplemental and published data set). I am not certain this statement is supported by the presented results. Certainly, the CFD shows changes with increased yaw angle, but there is a disconnect (something missing in the results/discussion) when it comes to comparing the stationary and mobile anemometer field data.

Figure 8 displays the difference between mean stationary and mobile anemometer measurements across each leg providing comparisons for head, tail and cross wind. A comparison of measurements coincident in time was not provided because of the difference in measurement frequency. Please see response to detailed comment 10c.

2 ) Section 2.1 – There are a number of terms presented (inlet velocity, yaw angle), coordinate systems (for the vehicle, anemometer, Earth), and instrument placement/height information that would be nice to see presented in a schematic. A simple top view of the truck, noting the location of the anemometer and identifying the coordinate systems that come into play would be very enlightening. This would also allow the authors to add lines/ reference marks that would show the locations of the cross sections used from the CFD results for Figures 3, and 4. As a reader I was very confused as to whether the cross section in Figure 3, for example ran down the centerline of the truck or was offset to the driver's side where the anemometer was located.

The following field photo and caption has been added to the supplementary material.

[Figure]

**Figure 1: Field photo showing the location of both anemometer placements. The North arrow indicates that the anemometers were positioned so the 0° reference was in line with the front of the truck.**

In the manuscript, Figure 3 corresponds to the longitudinal plane intersecting the tall anemometer, and Figure 4 corresponds to the lateral plane intersecting the tall anemometer.

3) When you developed the string model of the truck in figure 1, did you also include the roof rack and instrument mast in that model? As you note later, the mast can influence the observations and the FLUENT model should be able to assess the impact.

We did not include the roof rack and instrument mast in the ANSYS model. Our field experiment was planned to test multiple anemometer locations on the truck where-as our CFD simulations were designed to look at all of the differing anemometer locations in the same model for each yaw angle. We did not want to select an arbitrary location as placing the mast could skew the results of all locations.

4) Page 3, line 21 – Why did you choose exactly 25.4 m for the extent of the virtual wind tunnel? This seems a rather precise measurement. Why not 26 or 30 m. Did it relate to the analysis grid used by the CFD model.

The precision of this measurement came from scaling the truck model from imperial to metric units.

5) Page 4, line 6 – Can you comment on the impact/role of friction in your field experiment. How did the conditions differ from the smooth/ no slip conditions of the CFD.

The surface roughness on both the road and the vehicle differ from the smooth surfaces used in CFD. We selected the field placements to be mounted at a significant height above the vehicle to avoid direct comparison of CFD and field measurements that could be found close to the boundary layer of the vehicle.

6) Page 4, Line 8 – which "manufacturer's specifications were used? I assume this was ANSYS, but please specifically state that here as you have mentioned Toyata, Nissan, and other manufactures earlier in the paper.

This comment comes from the review of a paper that did not specify the manufacturer.

7) Page 5, ~line 8 – See my comment #2. Again, I think a schematic would make the paper easier to understand.

Please see response to comment #2.

8) CFD – general comment – It would be interesting to see the results if you turned the vehicle fully broadside (90°) to the wind (this is one of the worst angle for flow distortion on ships, even with bow mounted anemometers) and rotated to 180° to provide the tail wind example (this would likely be the worst case for tail wind flow, but may be enlightening). The lack of any CFD from the tail wind configuration gave you no standard for comparison for your tail wind field data.

We did not include the simulations of these scenarios in our comparison because the motion of the vehicle always created a frontal wind. A scenario where 100% of the airflow was coming from the vehicle was not representative of the conditions present in the field experiments, and would not provide a direct comparison. However, we agree that the results of both these scenarios would be interesting to see and could be used to further calibrate measurements collected in tail wind conditions.

9) Figure 2 – Please add a north reference to the map, and not how much difference between true north on the Earth (via your GPS data) and the roads on your grid. I assume they are not running exactly N/S or E/W.

A north reference has been added. The measurements were collected in Saskatchewan on township and range roads. The vehicle bearing measurements were calculated from the GPS latitude and longitude GPS coordinates, the roads used in the square were within 2° of North, South, East and West directions.

The following text has been added, "The calculated vehicle bearing measurements showed that the roads used in the square route were within 2° of North, South, East and West directions."

10) Page 6 – I would like to see a bit more regarding your experimental design here
a. You used two different Sonic anemometers (RM Young on the truck, Decagon on the stationary towers). What is the difference in their specifications? IF placed side by side, what is the bias you would expect between the two sensors.

If placed side by side we would expect to see a bias of approximately 1% in windspeed measurement and 1 degree in wind direction. The RM Young anemometer reports an accuracy of 2% for wind speed measurements below 30 m/s while decagon reports 3%. The RM Young anemometer reports an accuracy of 2 ° in wind direction while the Decagon reports 3°.

The Decagon anemometers were selected for stationary measurements because it was affordable to use multiple along the route. The RM Young anemometer was selected for the mobile measurements because it performs well under high speed conditions. The upper measurement limit for the Decagon anemometers is 30 m/s. We did not expect winds above 30 m/s to be present for the stationary measurements.

The following text has been added to Page 6 L9:
"The Decagon anemometer model was selected based on cost effectiveness, and the RM Young model was selected for its ability to measure high wind speeds, necessary for the mobile wind measurements."

b. What is the orientation of the anemometer on the truck and how was this confirmed? I assume the 0° line on the sonic was pointed towards the front of the truck, but this is not stated (nor is it essential as long as you account for an offset angle in the calculation of true winds – see Smith et al 1999). Again, this could be a schematic on the truck and the relevant coordinate systems.

The anemometer was orientated with the 0° line pointed towards the front of the truck. A ring was orientated on the mount to align the anemometer in this orientation. Also see response to comment #2.

c. Are the measurements from the Decagon averaged or instantaneous at the minute the values are reported.

The measurements from the Decagon are averaged at the minute they are reported. The text has been altered to read, "Measurements of time, wind speed, maximum gust, and direction were recorded with a Decagon Device Em50 datalogger. Averaged measurements were reported in one-minute intervals at the same time we were driving the route.

d. Were anemometers at the two heights on the truck run simultaneously, or were the positions swapped between test?

The anemometers were on the truck simultaneously.
P6L10 has been changed to, "To explore the effects of the anemometer height above the truck, we repeated tests with the addition of a secondary anemometer at a lower position of 0.4 m above the truck (2.1m above the ground), and 0.4 m from the longitudinal axis of the truck on the driver's side.

e. When you ran the test at a lower height, did you also lower the height of the stationary anemometers?

The stationary anemometer height was left the same for the double anemometer test.

11. Page 7, line 18-19 – This seems an expected result, and you should state that it was expected. Trucks are designed to be aerodynamic going into the wind as you drive down the road. Most manufacturers probably do not pay much attention to the aerodynamics in cross winds as this plays little role in fuel economy (though they may look at this some for safety specifications).

The following sentence has been added to P7 L19:
"This result was expected as vehicle's are designed to be aerodynamic in the forward direction of motion."

12. Figure 3 – In the caption (or the suggested schematic) please clarify the plane on which these CFD results are presented. I assume they are on a plane running the length of the truck that intersects the mast on the driver's side (not a plane down the center of the truck).

Please see response to comment #2.

13. Figure 4 – Same comment as Figure 3 (although here you do mention "along the roof racks, but the exact location of the rack vs the anemometer could be shown in a schematic or even a photograph taken in the field if you have one)

Please see response to comment #2.

14. Equation 2 – you note wind direction units are degrees, but you do not mention the coordinate system. Are these wind directions from the anemometer in the truck's coordinate system or do they need to be corrected to the Earth's coordinate system. Or does it not matter at all.

Good point, this equation refer's to the truck's coordinate system. The following text has been added to P10 L6:
"Equation 2 gives the side-mounted anemometer's correction function for wind direction (WD) measurements ranging from $-40° < WD < 40°$, where the wind direction measurement is measured in the truck's coordinate system with 0° facing the front of the truck."

15. Figure 6 – There is no explanation in the manuscript of how the synoptic conditions varied between your field tests. Were the days all very windy, calm, stormy, benign? Also does figure 6 show the comparison for just a single trial around your box for high and low anemometer positions of is this an average/ compilation of multiple circuits around your box?

The synoptic conditions varied. The following text has been added to P10 L 13:

"Vehicle-based wind measurements were acquired on separate days in predominantly North, East, South and West winds, and with the average wind speed of the field tests ranging from 13.5 to 27.1 km h-1."

Figure 6 shows the comparison of a single leg for high and low anemometer positions in cross wind.

16. Figure 7 – These are unacceptable for publication. They are too small to read and it is impossible to see the changes you try to address in the text.

The figure has been enlarged and sharpened.

17. Figure 8 – The contents of the tables need to be described in the figure caption. Also are the wind directions represented in the wind roses direction "from which" the wind is blowing versus the vectors from the stationary anemometer being an arrow pointing "to which" the wind blows? Further clarification is needed in the caption. For example, looking at the arrows rom the bottom leg of the box, it is not clear these are a tail wind without some additional explanation.

The following text has been added to the caption:
"The vehicle travelled this route in a clockwise direction. The wind roses show the direction from which the wind is blowing, the mapped vectors indicate the direction to which the wind is blowing. The table below each plot indicates the averaged stationary and mobile anemometer wind speed and wind direction for each leg."

18. Page 15, lines 12-19. This is really the big conclusion of the work and I think it could be stated more strongly. Based on your results and these previous studies, the conclusion is that an anemometer mounted on a truck should be placed "as far forward and a high as possible", with a location on a bumper mounted forward mast being the best location. This is exactly the guidance given to ship operators regarding wind measurement – forward and high is best. As you noted, if you mount on top of the truck anywhere behind the cab, you would need to correct the winds for flow distortion, but since the flow would

vary with each type of vehicle used and CFD modeling is expensive, a recommendation to go with a forward mounted (bumper) mast seems the best choice.

This comment is very true that this is a major conclusion. The following text has been added to P 15 L19 .

"Vehicle-based anemometer measurements should be calibrated for vehicle shape and anemometer placement but as this can be costly and time consuming truck-based anemometers should be placed as far forward and as high as possible to obtain the most accurate results."

19. Page 15, starting line 20 – This entire paragraph seems out of place and the paper would not be adversely affected if it was removed.

The author's have chosen to keep this paragraph as it can speak to how other mobile wind measurements have evolved over time.

20. Page 17, data availability – I downloaded the file with the anemometer measurements but found it to be lacking in documentation. There is some mention of units, etc. on the source web page, but I suggest taking the time to add another header line with the units for all the columns. For example, what are the time units (local or UTC)? And what is the EPOCH – a time stamp of some sort.

The dataset has been republished as v2 with a header row specifying the units and time zone. The data is available here: 10.6084/m9.figshare.9922274

[revised manuscript text omitted]